



# Leaf-scale experiments reveal important omission in the Penman-Monteith equation

Stanislaus J. Schymanski[1] and Dani Or[1]

[1]Department of Environmental Sciences, ETH Zurich, 8092 Zurich, Switzerland

*Correspondence to:* Stan Schymanski (stan.schymanski@env.ethz.ch)

**Abstract.** The Penman-Monteith (PM) equation is commonly considered the most advanced physically based approach to computing transpiration rates from plants considering stomatal conductance and atmospheric drivers. It has been widely evaluated at the canopy scale, where aerodynamic and canopy resistance to water vapour are difficult to estimate directly, leading to various empirical corrections when scaling from leaf to canopy. Here we evaluated the PM equation directly at the leaf scale, using a detailed leaf energy balance model and direct measurements in a controlled, insulated wind tunnel using artificial leaves with fixed and pre-defined "stomatal" conductance. Experimental results were consistent with a detailed leaf energy balance model; however, the results revealed systematic deviations from PM-predicted fluxes, which pointed to fundamental problems with the PM equation. Detailed analysis of the derivation by Monteith (1965) and later amendments revealed two errors in considering the effect of stomata and the two-sided exchange of sensible heat. A corrected set of analytical solutions for leaf temperature as well as latent and sensible heat flux is presented and comparison with the original PM equation indicates a major improvement in reproducing experimental results at the leaf scale. The errors in the original PM equation and its failure to reproduce experimental results at the leaf scale (for which it was originally derived) propagate into inaccurate sensitivities of transpiration and sensible heat fluxes to changes in atmospheric conditions, such as those associated with climate change (even with reasonable present day performance after calibration). The new formulation presented here rectifies some of the shortcomings of the PM equation and could provide a more robust starting point for canopy representation and climate change studies.

## 1  Introduction

The vast majority of current global land surface models, hydrological models and inverse approaches to deduce evaporation from remote sensing data employ the analytical solution for the latent heat flux from plant leaves derived by Monteith (1965), based on an earlier formulation for a wet surface by Penman (1948). This so-called Penman-Monteith equation (henceforth referred to as the PM equation), which introduced stomatal resistance into Penman's formalism, found widespread use in the prediction of latent heat flux based on estimates of leaf and canopy resistance to water vapour. Whereas the PM equation is generally believed to provide an adequate physical description of transpiration from an individual leaf, it is commonly applied at the canopy scale, where aerodynamic and bulk stomatal resistance are difficult to estimate and are usually deduced empirically from measurements of transpiration and an inverted PM equation (Raupach and Finnigan, 1988) or from observed





surface temperatures (Tanner and Fuchs, 1968). The scaling up from leaf to canopy and use of data at daily or monthly scales has led to various empirical corrections to the PM equation (Allen, 1986; Langensiepen et al., 2009), which may have obscured more fundamental issues with the derivations by Monteith (1965). A number of authors have focused on biases introduced by the simplifications inherent in the PM equation, such as the linearisation of the saturation vapour pressure curve and the

neglect of dependency of net irradiance on surface temperature, and proposed various approaches to reduce such biases (Paw U and Gao, 1988; McArthur, 1990; Milly, 1991; Widmoser, 2009). Interestingly, even 50 years after its derivation, we have not found a rigorous test of the PM equation at the leaf scale, whereas our analysis of the derivations by Monteith (1965) and later amendments revealed two errors in considering the effect of stomata and the two-sided exchange of sensible heat.

Therefore, the objectives of the present study are to (1) develop an experimental setup allowing direct and independent

measurement of all components of the energy balance of a single leaf and the relevant boundary conditions, (2) compare different analytical and numerical leaf energy balance and gas exchange models with experimental results, and (3) derive an improved analytical representation of latent and sensible heat fluxes at the leaf scale.

The study is structured as follows. We first present a physically-based, explicit leaf energy balance and gas exchange model, to serve as a reference for the physical processes. The explicit model is then used to re-derive the Penman and Penman-

Monteith (PM) equations while highlighting all simplifying assumptions inherent in these formulations. Subsequently, we will derive a general analytical formulation based on the approach by Penman (1952) and analyse consistency between the various analytical solutions and the explicit leaf energy and gas exchange model. In the next step, we will present an experimental setup allowing to measure all components of the leaf energy balance under fully controlled conditions, using artificial leaves with known stomatal conductance. Experimental results will be compared with the explicit numerical model and the different

analytical solutions, assessing potential bias.

## 2  Materials and Methods

The detailed derivations are described in the appendix, while the experimental methods will be discussed in detail in a technical note to be submitted to HESS (Schymanski et al., in prep.). Here, we only summarise the key points and concepts necessary to understand the flow of the paper. All symbols used in this paper are listed and described in the appendix, Tables A1 and A2.

### 2.1  Explicit leaf energy balance and gas exchange model

The detailed leaf energy balance model used here is based on derivations published previously (Schymanski et al., 2013; Schymanski and Or, 2015, 2016), and is reproduced here after re-organisation of equations for consistency with the present paper.

The leaf energy balance is determined by the dominant energy fluxes between the leaf and its surroundings, including

radiative, sensible, and latent energy exchange (linked to mass exchange). These are illustrated in Fig. 1. Focusing on steady-state conditions, the energy balance can be written as:

$$R_s = R_{ll} + H_l + E_l, \tag{1}$$





where $R_s$ is absorbed short-wave radiation, $R_{ll}$ is the net emitted long-wave radiation, i.e. the emitted minus the absorbed, $H_l$ is the sensible heat flux away from the leaf and $E_l$ is the latent heat flux away from the leaf. In the above, extensive variables are defined per unit leaf area. Following our previous work (Schymanski et al., 2013), this study considers spatially homogeneous planar leaves, i.e. homogenous illumination and a negligible temperature gradient between the two sides of the leaf. The net longwave emission is represented by the difference between blackbody radiation at leaf temperature ($T_l$) and that at the temperature of the surrounding objects ($T_w$, commonly represented by air temperature, $T_a$) (Monteith and Unsworth, 2007):

$$R_{ll} = a_{sH} \epsilon_l \sigma (T_l^4 - T_w^4), \tag{2}$$

where $a_{sH}$ is the fraction of projected leaf area exchanging radiative and sensible heat (2 for a planar leaf, 1 for a soil surface), $\epsilon_l$ is the leaf's longwave emmissivity ($\approx 1$) and $\sigma$ is the Stefan-Boltzmann constant. Total convective heat transport away from the leaf is represented as:

$$H_l = a_{sH} h_c (T_l - T_a), \tag{3}$$

where $h_c$ is the average one-sided convective heat transfer coefficient, determined by properties of the leaf boundary layer.

Latent heat flux ($E_l$, W m$^{-2}$) is directly related to the transpiration rate ($E_{l,mol}$) by:

$$E_l = E_{l,mol} M_w \lambda_E, \tag{4}$$

where $M_w$ is the molar mass of water and $\lambda_E$ the latent heat of vaporisation. $E_{l,mol}$ (mol m$^{-2}$ s$^{-1}$) was computed in molar units as a function of the concentration of water vapour within the leaf ($C_{wl}$, mol m$^{-3}$) and in the free air ($C_{wa}$, mol m$^{-3}$) (Incropera et al., 2006, Eq. 6.8):

$$E_{l,mol} = g_{tw} (C_{wl} - C_{wa}), \tag{5}$$

where $g_{tw}$ (m s$^{-1}$) is the total leaf conductance for water vapour, dependent on stomatal ($g_{sw}$) and boundary layer conductance ($g_{bw}$) in the following way:

$$g_{tw} = \frac{1}{\frac{1}{g_{sw}} + \frac{1}{g_{bw}}} \tag{6}$$

The leaf boundary layer conductances to sensible heat and water vapour ($h_c$ and $g_{bw}$ respectively) depend on leaf size ($L_l$), wind speed ($v_w$) and the level of turbulence in the air stream ($N_{Re_c}$), expressed in the dimensionless Nusselt and Lewis numbers ($N_{Nu_L}$ and $N_{Le}$ respectively). The relation of $h_c$ to $g_{bw}$ additionally depends on whether stomata are present on one side of the leaf only ($a_s = 1$) or both sides of the leaf ($a_s = 2$). The relevant equation to compute all of these variables as a function of air temperature, pressure and vapour pressure ($T_a$, $P_a$ and $P_{wa}$ respectively), wind speed ($v_w$), turbulence and leaf properties are given in the Appendix, Sections B1–B4.

Figure 2 illustrates the use of measurements and the different equations to compute the leaf energy balance components. Leaf temperature ($T_l$) needs to be computed by iteration, using the leaf energy balance model, due to the non-linearities in Eq.





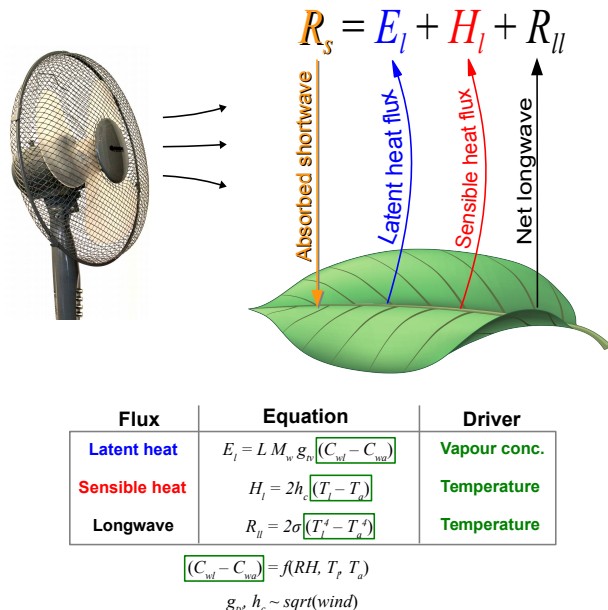

**Figure 1.** Components of the leaf energy balance and their thermodynamic drivers. Bent arrows indicate fluxes that are directly affected by wind speed. Table at bottom illustrates the drivers for each flux (temperature differences for sensible and radiative heat exchange, water vapour concentration differences for mass exchange and hence latent heat flux). Additional equations below the table illustrate that the driver for latent heat flux is also related to temperature differences and that the transfer coefficients for both latent and sensible heat flux depend on wind. $L$: latent heat of vaporisation, $M_w$: molecular mass of water, $g_{tw}$ total leaf conductance to water vapour, $C_{wl}$: concentration of water vapour in leaf-internal air, $C_{wa}$: concentration of water vapour in free air stream, $h_c$: one-sided heat transfer coefficient, $T_l$: leaf temperature, $T_a$: air temperature, $\sigma$: Stefan-Boltzmann constant, $RH$: relative humidity of the free air stream. $g_{bw}$: leaf boundary layer conductance to water vapour.

2 and Eq. B5. Note that a direct measurement of $T_l$ (e.g. using infrared sensors) would enable direct computation of $R_{ll}$ and $H_l$, and finally $E_l$ from the energy balance as $E_l = R_s - R_{ll} - H_l$ without any iterations. This illustrates that the use of any of the analytical solutions explained below is not necessary if $T_l$ is known, and questions the approach proposed by Tanner and Fuchs (1968), where observed leaf or surface temperature is inserted into the Penman-Monteith equation to estimate transpiration rate.

## 2.2 Generalisation of Penman's analytical approach

The PM-equation derived by Monteith (1965) was based on the analytical solution for evaporation from a wet surface by Penman (1948). The key point of Penman's analytical solution is to express evaporation as a function of the surface-air vapour pressure difference and sensible heat flux as a function of surface-air temperature difference. Here we will follow the succinct derivation presented in the appendix of Penman (1952) and use our notation for a leaf to obtain a general solution applicable





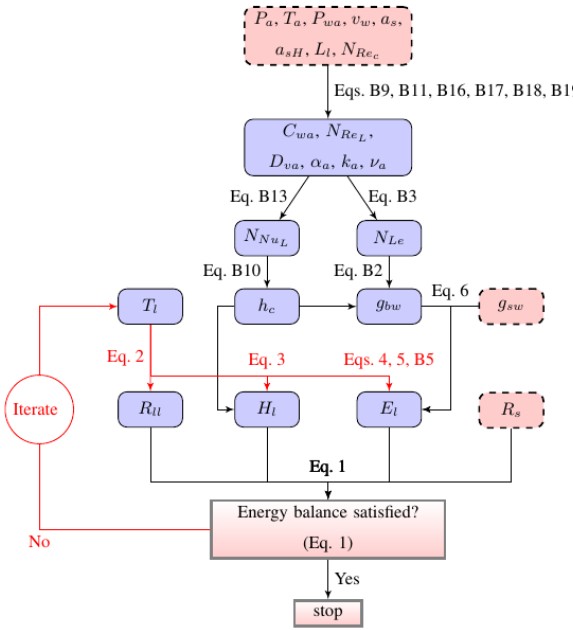

**Figure 2.** Flow chart of computation procedure for different leaf energy balance components. Dashed, pink boxes with rounded corners indicate external input, while solid, blue rounded boxes indicate computed variables. Note the central role of leaf temperature, which needs to be computed by iteration against the leaf energy balance.

both to a transpiring leaf or an evaporating surface. In the first step, we will introduce general transfer coefficients for latent heat ($c_E$, W m$^{-2}$ Pa$^{-1}$) and sensible heat ($c_H$, W m$^{-2}$ K$^{-1}$), satisfying the following equations:

$$E_l = c_E(P_{wl} - P_{wa}) \tag{7}$$

and

$$H_l = c_H(T_l - T_a) \tag{8}$$

(Please refer to Appendix B3 for a discussion of the meaning of Eq. 7 compared to Eq. 5, and conversion of transfer coefficients.)

Eqs. 7, 8 and the leaf energy balance equation (Eq. 1), form a system of three equations with four unknowns: $E_l$, $H_l$, $T_l$ and $P_{wl}$. In order to eliminate $T_l$, Penman assumed that the ratio of the vapour pressure difference between the surface and the saturation vapour pressure at air temperature ($P_{was}$) to the temperature difference between the surface and the air can be approximated by the slope of the saturation vapour pressure curve at air temperature ($\Delta_{eTa}$):

$$\Delta_{eTa} = \frac{P_{wl} - P_{was}}{T_l - T_a} \tag{9}$$





This gives four equations (Eqs. 1, 7, 8, and 9) that can be solved for the four unknowns $E_l$, $H_l$, $T_l$ and $P_{wl}$:

$$E_l = \frac{\Delta_{eTa}c_E(R_s - R_{ll}) + c_Ec_H(P_{was} - P_{wa})}{\Delta_{eTa}c_E + c_H}, \tag{10}$$

$$H_l = \frac{c_H(R_s - R_{ll}) + c_Ec_H(P_{wa} - P_{was})}{\Delta_{eTa}c_E + c_H}, \tag{11}$$

$$T_l = T_a + \frac{(R_s - R_{ll}) + c_E(P_{wa} - P_{was})}{\Delta_{eTa}c_E + c_H} \tag{12}$$

and

$$P_{wl} = \frac{\Delta_{eTa}(R_s - R_{ll} + P_{wa}c_E) + P_{was}c_H}{\Delta_{eTa}c_E + c_H} \tag{13}$$

In the original formulations by Penman and Monteith, the term $R_s - R_{ll}$ is referred to as net available energy, and for a ground surface, it is represented by net radiation minus ground heat flux ($R_N - G$). For a leaf, there is no ground heat flux, and $R_N = R_s - R_{ll}$. In most applications of the analytical solutions, $R_{ll}$ is not explicitly calculated, but it is assumed that $R_N$ is known, neglecting the dependence of $R_{ll}$ on the leaf temperature. This neglect can be alleviated by linearising the equation for

$R_{ll}$ (Leuning et al., 1989), which was also done in Section 2.4, where we re-derive Eqs. 10–12 based on a linearised equation for $R_{ll}$, eliminating the need for separate estimation of $R_N$.

To solve Eqs. 10–13, one only needs information about $c_H$ and $c_E$, appropriate for a leaf or an evaporating surface, whichever is the system of interest. For a planar leaf, $c_H = a_{sH}h_c$ with $a_{sH} = 2$ as the leaf exchanges sensible heat on both sides, whereas for a soil surface, $a_{sH} = 1$. Comparison of Eqs. 4 and 7 with the common representation of $E_{l,mol}$ as a function of total leaf

conductance to water vapour ($g_{tw}$) and water vapour mole fractions (Eq. B6) suggests that

$$c_E = M_w\lambda_E g_{tw,mol}/P_a, \tag{14}$$

where $g_{tw,mol}$ has an aerodynamic component related to $g_{bw}$ (and hence $h_c$) and a surface-specific component, related to $g_{sw}$, as described in Appendix B1. Since planar leaves can have stomata on one or both sides, the relation between $h_c$ and $g_{bw}$ is not universal, i.e. $a_s$ in Eq. B2 can be equal to 1 or 2, whereas for a soil surface $a_s = 1$.

## 2.3  Inconsistencies in the PM equation

From the general form (Eqs. 10–12), we can recover various analytical forms used for latent heat flux (e.g. Penman, 1948, 1952; Monteith, 1965), with the appropriate substitutions for $c_E$ and $c_H$. This is shown in detail in the appendix, Section B8, where we also illustrate some inconsistencies in the published derivations. Here, we will discuss errors in the derivation of the PM-equation, when intended for the simulation of leaf transpiration. The derivation is based on the Penman equation for a wet

surface (Penman, 1948), which can be recovered from the above general solution by substituting $c_E = f_u$ and $c_H = \gamma_v f_u$ into Eq. 10 (Fig. 3a):

$$E_w = \frac{\Delta_{eTa}(R_s - R_{ll}) + f_u\gamma_v(P_{was} - P_{wa})}{\Delta_{eTa} + \gamma_v}, \tag{15}$$





where $f_u$ is usually referred to as the "wind function".

Monteith (1965) re-derived the Penman equation for wet surface evaporation (Eq. 15) using a different set of arguments and arrived to an equivalent equation (Eq. 8 in Monteith (1965)):

$$E_w = \frac{\Delta_{eTa}(R_s - R_{ll}) + \rho_a c_{pa}(P_{was} - P_{wa})/r_a}{\Delta_{eTa} + \gamma_v}, \tag{16}$$

where $r_a$ is the leaf boundary layer resistance to sensible heat flux. Eq. 16 is consistent with Eq. 15 if Penman's wind function ($f_u$) is replaced by:

$$f_u = \frac{\rho_a c_{pa}}{\gamma_v r_a}. \tag{17}$$

Monteith pointed out that the ratio between the conductance to sensible heat and the conductance to water vapour transfer, expressed in the psychrometric constant ($\gamma_v$) would be affected by stomatal resistance ($r_s$) and hence proposed to replace the psychrometric constant by $\gamma_v^*$:

$$\gamma_v^* = \gamma_v(1 + \frac{r_s}{r_a}), \tag{18}$$

leading to the so-called Penman-Monteith equation for transpiration:

$$E_l = \frac{\Delta_{eTa}(R_s - R_{ll}) + \rho_a c_{pa}(P_{was} - P_{wa})/r_a}{\Delta_{eTa} + \gamma_v\left(1 + \frac{r_s}{r_a}\right)} \tag{19}$$

Eq. 19, with $\gamma_v$ defined in Eq. B46, could be recovered by substituting $c_E = \epsilon \lambda_E \rho_a / (P_a(r_s + r_v))$ and $c_H = c_{pa}\rho_a/r_a$ into Eq. 10, with subsequent substitution of $r_v = r_a$ (implicit in Eq. 17, considering that $f_u = c_E$). Note, however, that $r_a$ in Monteith's derivation is defined as one-sided resistance to sensible heat exchange (Monteith and Unsworth, 2013, P. 231), neglecting the fact that planar leaves exchange sensible heat on both sides. We suppose that this omission is related to the original Penman derivation, developed for a soil surface, which exchanges latent and sensible heat across one interface, and hence is not appropriate for a leaf. To alleviate this constraint, one could define $r_a$ and $r_s$ as total (two-sided) leaf resistances, but in this case, the simplification $r_v \approx r_a$ is not valid for hypostomatous leaves, as $r_v$ would then be twice the value of $r_a$. This is illustrated in Fig. 3c, where sensible heat flux is released from both sides of the leaf, while latent heat flux is only released from the abaxial side, implying that $a_{sh} = 2$ and $a_s = 1$.

Monteith and Unsworth (2013) acknowledged that a hypostomatous leaf could exchange sensible heat on two sides, but latent heat on one side only and proposed to represent this fact by further modifying $\gamma_v^*$ to:

$$\gamma_v^* = n_{MU}\gamma_v(1 + r_s/r_a) \tag{20}$$

where $n_{MU} = 1$ for leaves with stomata on both sides and $n_{MU} = 2$ for leaves with stomata on one side, i.e. $n_{MU} = a_{sh}/a_s$ in our notation. Insertion of Eq. 20 into Eq. 16 yields what we will call the Monteith-Unsworth (MU) equation, which only differs from the Penman-Monteith equation by the additional factor $n_{MU}$:

$$E_l = \frac{\Delta_{eTa}(R_s - R_{ll}) + \rho_a c_{pa}(P_{was} - P_{wa})/r_a}{\Delta_{eTa} + \gamma_v n_{MU}\left(1 + \frac{r_s}{r_a}\right)} \tag{21}$$





However, this was done by specifying $r_s$ and $r_a$ as one-sided resistances when inserting them into the term for $\gamma_v$ in Eq. 16, which was already based on the approximation $r_v \approx r_a$, which is not valid for hypostomatous leaves, as explained above. If we replace $r_a$ by $r_a = r_a/a_{sh}$ in Eq. 16 *before* substitution of Eq. 20, we obtain a corrected MU-equation:

$$E_l = \frac{\Delta_{eTa}(R_s - R_{ll}) + \rho_a c_{pa}(P_{was} - P_{wa})a_{sh}/r_a}{\Delta_{eTa} + \gamma_v a_{sh}/a_s \left(1 + \frac{r_s}{r_a}\right)}, \tag{22}$$

which only differs from Eq. 21 by the factor $a_{sh}$ ($= 2$) in the nominator. Eqs. 19 and 22 are only equivalent to each other if $a_{sh} = 1 = a_s$, implying that Eq. 19 is not applicable for any planar leaves. For symmetrical amphistomatous leaves, $a_{sh} = 2 = a_s$, in which case the classic PM equation is only missing a factor of 2 in the nominator, as pointed out by Jarvis and McNaughton (1986, Eq. A9).

## 2.4 Analytical solution including radiative feedback

The above analytical solutions eliminated the non-linearity problem of the saturation vapour pressure curve, but they do not consider the dependency of the longwave component of the leaf energy balance ($R_{ll}$) on leaf temperature ($T_l$), as expressed in Eq. 2. Therefore, the above analytical equations are commonly used in conjunction with fixed value of $R_{ll}$, either taken from observations or the assumption that $R_{ll} = 0$. Here we replace the non-linear Eq. 2 by its tangent at $T_l = T_a$, which is given by:

$$R_{ll} = 4a_{sh}\epsilon_l\sigma T_a^3 T_l - a_{sh}\epsilon_l\sigma(T_w^4 + 3T_a^4) \tag{23}$$

Note that the common approximation of $T_w = T_a$ simplifies the above equation to $R_{ll} = 4a_{sh}\epsilon_l\sigma(T_a^3 T_l - T_a^4)$. The linearisation introduces a bias of less than -20 W m$^{-2}$ in the calculation of $R_{ll}$ for leaf temperatures $\pm 20$ K of air temperture, compared to Eq. 2 (see Fig. A3).

We can now use a similar procedure as in Section 2.2, but this time aimed at eliminating $P_{wl}$ using the Penman assumption, rather than eliminating $T_l$. We first eliminate $c_E$ from Eq. 7 by introducing the psychrometric constant as

$$\gamma_v = c_H/c_E \tag{24}$$

and introduce it into Eq. 8 to obtain:

$$H_l = \gamma_v c_E(T_l - T_a) \tag{25}$$

Then, we insert the Penman assumption (Eq. 9) to eliminate $P_{wl}$ and obtain:

$$E_l = \frac{c_H\left(\Delta_{eTa}(T_l - T_a) + P_{was} - P_{wa}\right)}{\gamma_v} \tag{26}$$

We can now insert the linearised Eq. 23, Eq. 26 and Eq. 8 into the energy balance equation (Eq. 1), and solve for leaf temperature ($T_l$) to obtain:

$$T_l = \left( R_s + c_H T_a + c_E\left(\Delta_{eTa}T_a + P_{wa} - P_{was}\right) \right.$$
$$\left. + a_{sh}\epsilon_l\sigma\left(3T_a^4 + T_w^4\right)\right) \frac{1}{c_H + \Delta_{eTa} + 4a_{sh}\epsilon_l\sigma T_a^3} \tag{27}$$

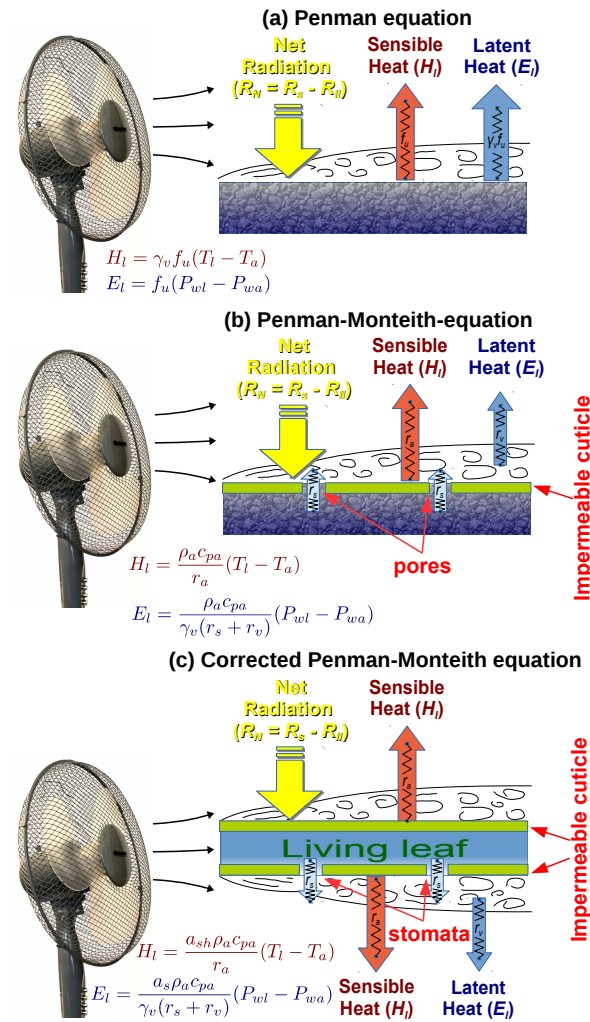

**Figure 3.** Different representations of energy partitioning into sensible and latent heat flux. (a) Penman equation, where net radiation is partitioned between ground heat flux (not shown), sensible heat flux and latent heat flux at the land surface, affected by boundary layer resistance expressed in wind function ($f_u$); (b) Penman-Monteith equation, considering additional stomatal resistance ($r_s$); and (c) corrected Penman-Monteith equation for a hypo-stomatous leaf, where sensible heat flux is emitted from both sides of the leaf ($a_{sh} = 2$), while latent heat flux is only released on the abaxial (lower) side of the leaf ($a_s = 1$).

where the temperature of the surroundings is commonly assumed to equal air temperature ($T_w = T_a$). Eq. 27 can be re-inserted
into Eqs. 8, 26 and 23 to obtain analytical expressions for $H_l$, $E_l$ and $R_{ll}$ respectively, which satisfy the energy balance (Eq. 1). Alternatively, the value of $T_l$ obtained from Eq. 27 for specific conditions could be used to calculate any of the energy balance components using the fundamental equations described in Fig. 2. However, in this case, bias in $T_l$ due to simplifying assumptions included in the derivation of Eq. 27 could result in an unclosed leaf energy balance ($R_s - R_{ll} - H_l - E_l \neq 0$).





## 3 Experimental setup

To separate the physical aspects of leaf energy and gas exchange from complex biological control, we used artificial leaves with laser-perforated surfaces representing fixed stomatal apertures and continuous water supply monitored by micro flow sensors (Fig. 4). We further constructed a specialised insulated leaf wind tunnel permitting full control atmospheric conditions including air temperature, humidity, irradiance and wind speed and allowing direct measurement of all leaf energy balance components independently, including net radiation latent and sensible heat flux. A detailed documentation of the leaf wind

tunnel and the artificial leaves along with the relevant thermodynamic calculations will be submitted as a technical note to HESS (Schymanski and Or, in prep.).

### 3.1  Artificial leaves

The artificial leaves were constructed of a core made of porous filter paper (Whatman No. 41), glued onto aluminium tape and connected to a water supply by a thin tube, flattened at one end and tightly glued between the aluminium foil and the filter paper,

using Araldite epoxy resin (Fig. 4). Along with the water supply tube, a thin copper-constantan thermocouple (TG-TI-40) was placed between the filter paper and the adhesive aluminium tape. The water supply was connected to a high resolution liquid flow meter (SLI-0430, Sensirion AG, Staefa, Switzerland) and a water supply with a water table 1-3 cm below the position of the leaf, to ensure that the liquid flow did not exceed the transpiration rate while maintaining minimum head loss along the flow path.

Different laser perforations were performed by Ralph Beglinger (Lasergraph AG, Würenlingen, Switzerland), Robert Voss (ETH Zurich, Switzerland) and Rolf Brönnimann (EMPA, Zurich, Switzerland) and the geometry of laser perforations was measured using a confocal laser scanning microscope (CLSM VK-X200, Keyence, Osaka, Japan). See Fig. A2 for examples.

The stomatal conductance resulting from a particular perforation size and density was computed following the derivations presented by Lehmann and Or (2015), assuming that the stomatal conductance results from two resistances in series, the

throat resistance ($r_{sp}$), resulting from the width of the perforation and the thickness of the perforated foil, and the vapour shell resistance ($r_{vs}$), resulting from the size and spacing of the stomata, which can be understood as the resistance related to distribution of the point source water vapour over the entire one-sided leaf boundary layer. We hereby neglect any internal resistance (termed "end correction" by Lehmann and Or (2015)), as we assume that the wet filter paper has direct contact with the perforated foil. The relevant equations are described in Appendix B10.

### 3.2  Leaf wind tunnel

Leaf energy and gas exchange were measured in a thermally insulated wind tunnel with full control over energy and mass exchange (Fig. 4). The wind tunnel is circular, with two straight sections of 25 cm length each, a fan in one of the straight sections and a transparent window and leaf holder in the opposite straight channel. The fan circulates the wind as indicated by the arrows in Fig. 4, subjecting it to controlled wind conditions. The wind tunnel features an air inlet just before the fan and

an air outlet just after the fan, where the air is assumed to be well mixed across the tunnel cross-section. In this way, leaf gas





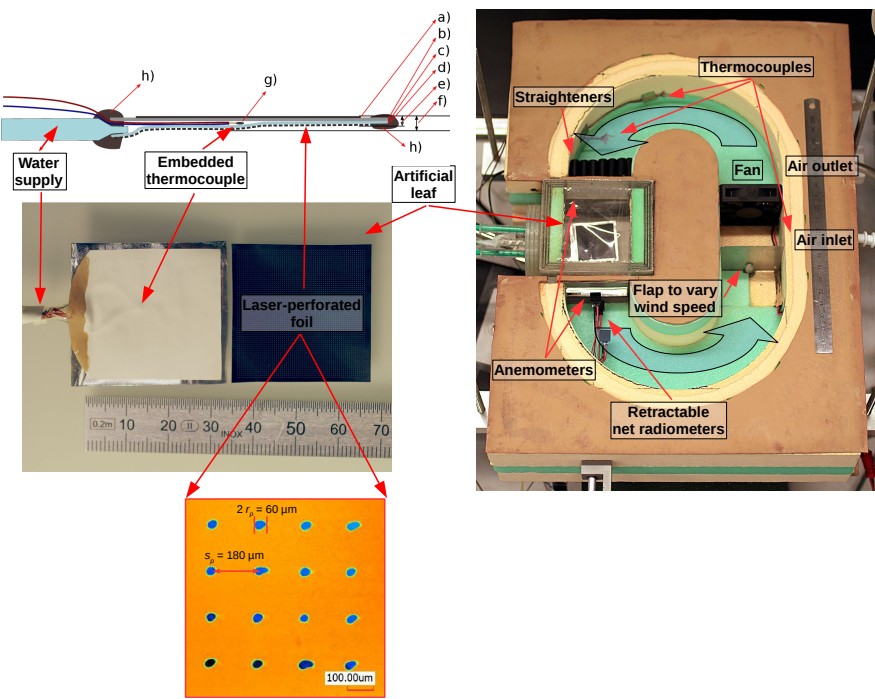

**Figure 4.** Artificial leaf and wind tunnel. Top left: cross-section of artifical leaf; center left: leaf image before full assembly; bottom left: topography of laser-perforated foil with 60 $\mu$m pore diameter and 180 $\mu$m spacing; right: wind tunnel. a) black aluminium tape (0.05 mm thick); b) aluminium tape (0.08 mm); c) absorbent filter paper (0.1-0.2 mm); d) laser-perforated foil (0.01-0.05 mm); e) min. leaf thickness: 0.3-0.4 mm; f) max. leaf thickness: 0.35-0.65 mm; g) thermocouple; h) glue; i) water supply tube (from flow meter).

exchange can be deduced from the concentration difference between the incoming and outgoing air and the controlled flow rate of air into the wind tunnel. For this purpose, the incoming air was supplied by a humidifier providing prescribed vapour pressure and flow rate.

The sensible heat flux ($H_l$) was deduced from the chamber energy balance, by computing the amount of heat exchanged
with the surroundings through the exchange of air and subtracting the amount of heat added by the fan. Since the fan was placed inside the chamber, the amount of heat it injected was assumed to be equal to its power consumption, which was kept constant by a programmable power controller, while wind speed was varied through adjusting the position of a wing in the flow path (Fig. 4) and monitored using a miniature thermal flow sensor. A stack of 3 cm long plastic straws in the flow path was used to reduce spiralling of the air flow caused by the rotating fan. The main wind tunnel was built of foamed insulation
material, while the leaf chamber itself had two layers of polypropylen foil on each side (above and below the leaf) to permit the transmission of shortwave and longwave radiation while minimising conductive heat transfer (see position of the artificial leaf in Fig. 4). We used retractable miniature net radiation sensors to periodically measure the net radiative load on the leaf.





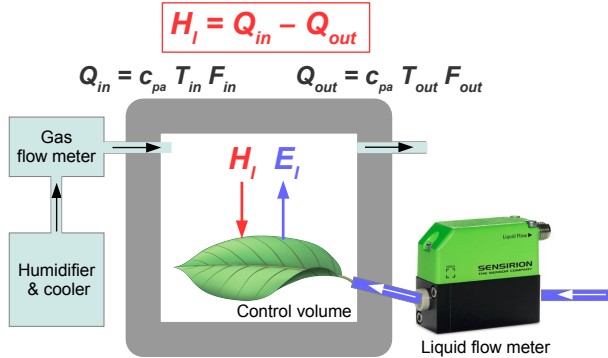

**Figure 5.** Simplified energy balance of insulated wind tunnel. Latent heat flux ($E_l$) is calculated from liquid flow rate into leaf, sensible heat flux ($H_l$) is calculated from difference in heat content of incoming and outoing air ($c_{pa}$: heat capacity of air; $T_{in}, T_{out}$ air temperatures of incoming and outgoing air; $F_{in}, F_{out}$: incoming and outgoing air flow rates).

Copper-constant thermocouples were placed in the air stream upstream and downstream of the leaf chamber, lightly inserted into the wind tunnel wall on the inside and the outside of the chamber, and in the duct through which air was supplied to the wind tunnel by an external humidifier providing a flow rate of up to 10 l/min and controlled air temperature and dew point.

The leaf wind tunnel was used to measure steady state conditions under given forcing (air temperature, humidity, wind speed and irradiance). Sensible heat exchange between the leaf and the surrounding air was computed from total chamber heat exchange, using monitored flow rate and temperature of incoming and outgoing air (Fig. 5). The relevant thermodynamic calculations will be presented in a separate technical note (Schymanski and Or, in prep.).

## 4  Results

### 4.1  Capacity of different formulations to reproduce experimental results using controlled conditions and artificial leaves

Experiments were performed for various artificial leaves with different stomatal conductances under varying air humidity or varying wind speed, in the absence of shortwave radiation. Stomatal conductance was deduced form confocal laser scanning microscope (CLSM) images of the perforated foils, as described above. The ranges of stomatal geometries and deduced conductances for the two different leaves presented here are given in Table 1. A more detailed analysis of the whole set of experimental results will be presented in a technical note (Schymanski and Or, in prep.). Here we only report two experiments under varying vapour pressure, which illustrate the general behaviour we found. Variations in leaf temperature and the various leaf energy balance components were simulated using the detailed numerical model and a simplified one by Ball et al. (1988), as well as different analytical solutions, including the Penman-Monteith equation ("PM", Eq. 19), the Monteith-Unsworth equation ("MU", Eq. 21), our corrected Monteith-Unsworth equation ("MUc", Eq. 22) and the analytical solution using linearised net



**Table 1.** Perforation characteristics and resulting stomatal conductances. Foil thickness: 25 $\mu$m.

| Pore density mm$^{-2}$ | Pore area $\mu$m$^{-2}$ | Pore radius $\mu$m | $g_{sw}$ m s$^{-1}$ |
|---|---|---|---|
| 31.2–36.4 | 1414–2317 | 21–27 | 0.027–0.042 |
| 7.8 | 1604–1932 | 22.5–24.8 | 0.0074–0.0086 |

$g_{sw}$: stomatal conductance

longwave balance ("Rlin", based on Eq. 27). The results obtained using the numerical model based on Ball et al. (1988) were very similar to the detailed numerical model presented here and were hence left out of the plots for clarity.

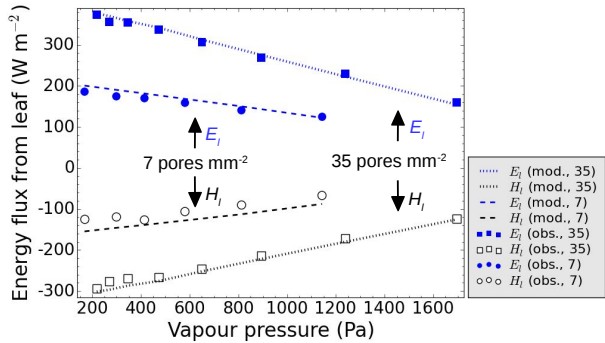

**Figure 6.** Numerical simulations vs. observed fluxes of sensible and latent heat in response to varying vapour pressure. Numerical model results (lines) based on observed boundary conditions representative of observations (dots). Labels of 35 and 7 pores mm$^{-2}$ correspond to the first and the second entry respectively in Tab. 1. The boundary conditions are summarised as follows. 35 perforations per mm$^2$: $g_{sw} = 0.035$ m s$^{-1}$; $R_s = 0$; $T_a = 295.7 – 296.0$ K; $v_w = 1.0$ m s$^{-1}$. 7.8 perforations per mm$^2$: $g_{sw} = 0.0074$ m s$^{-1}$; $R_s = 0$; $T_a = 296.1–296.7$ K; $v_w = 0.7$ m s$^{-1}$ $E_l$: latent heat flux; $H_l$: sensible heat flux.

The numerical model reproduced observed sensible and latent heat fluxes very accurately (Fig. 6) using stomatal conductance
values within the narrow ranges deduced from CLSM images (Tab. 1) with no other forms of calibration. The experimental conditions and stomatal conductances are given in the figure caption.

The analytical models generally under-estimated latent heat flux, but the model based on linearised $R_{ll}$ ("Rlin") showed very little bias and closely reproduced the observed latent and sensible heat fluxes, as it permitted calculation of the net longwave component (in contrast with PM, MU and MUc expressions that assumed $R_{ll} = 0$). The calculations based on the Penman-
270 Monteith equation significantly under-estimated latent heat flux, especially at high stomatal conductances (simulated values less than half of the observed in Fig. 7). The Monteith-Unsworth (MU) equation produced an even stronger under-estimation of latent heat flux in our results, whereas our corrected Monteith-Unsworth (MUc) equation was a lot closer to the observed heat fluxes than either the MU or the PM equations. However, only Eq. 27 (Rlin) was able to capture the asymmetry between



latent and sensible heat fluxes caused by net absorption of longwave radiation, as all the other calculations were based on the
assumption of zero radiative exchange ($R_{ll} = 0$), i.e. $H_l = -E_l$.

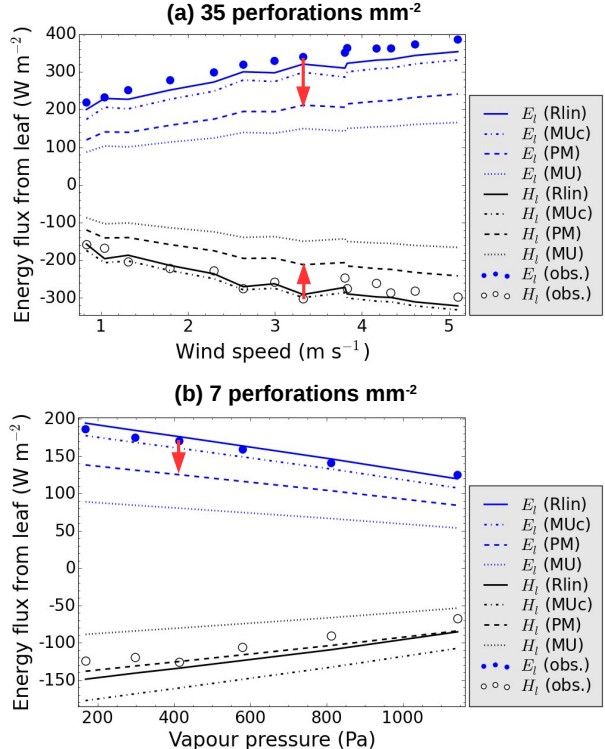

**Figure 7.** Analytical simulations vs. observed fluxes of sensible and latent heat in response to varying vapour pressure. Numerical model results (lines) based on observed boundary conditions representative of observations (dots). Conditions same as in Fig. 6. $E_l$: latent heat flux; $H_l$: sensible heat flux; "Rlin": based on linearised longwave balance (Eq. 27); "MUc": corrected Monteith-Unsworth equation (Eq. 22); "PM": Penman-Monteith equation (Eq. 19); "MU": Monteith-Unsworth equation (Eq. 21). Red arrows indicate the magnitudes of biases in the PM equation.

Since we were not able to systematically assess the effects of irradiance and air temperature in our lab experiments, we conducted a numerical experiment, where we compared simulations by the numerical model with simulations by the best analytical model and the PM-equation. The results shown in Fig. 8 suggest that our new analytical solution (Eq. 27) behaves very similarly to the numerical model, whereas the PM-equation misrepresents the sensitivities of latent and sensible heat fluxes to both irradiance and air temperature.





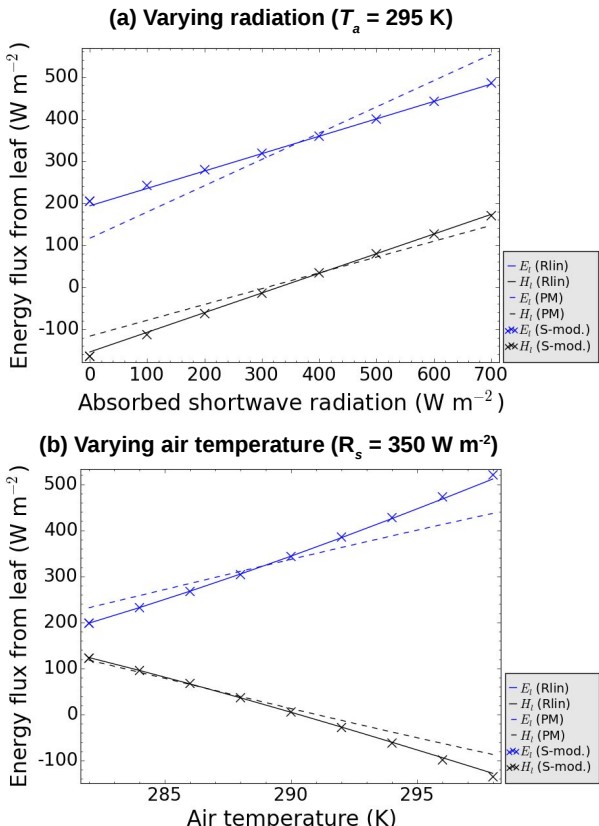

**Figure 8.** Numerical vs. analytical simulations of sensible and latent heat in response to varying irradiance and air temperature. Crosses represent numerical solution of leaf energy balance model ('S-mod.'), solid lines our new analytical solution ('Rlin') and dashed lines the Penman-Monteith equation. Simulation conditions: $g_{sw} = 0.045$ m s$^{-1}$); $P_{wa} = 1300$ Pa; $v_w = 1$ m s$^{-1}$. $E_l$: latent heat flux; $H_l$: sensible heat flux; "Rlin": based on linearised longwave balance (Eq. 27); "PM": Penman-Monteith equation (Eq. 19).

## 5 Discussion

"This age values usefulness more highly than correctness, and the making of money more highly than both. In fact, there is definitely something suspect about an examiner who would bother at all with whether an idea is correct or not." (Raupach and Finnigan, 1988)

The widespread use of the PM equation is mainly due to its simplicity and usefulness, the latter of which is contingent on its ability to accurately represent the sensitivity of evapotranspiration to atmospheric variables and surface properties (boundary layer and bulk stomatal conductances).

In our mathematical analysis, we found two errors in the PM equation as well as in the "corrected" formulation by Monteith and Unsworth (2013) related to the consideration of single-sided evaporating soil surface when deriving the equations. A





leaf exchanges sensible heat and longwave radiation from its two sites, whereas a soil has only one side exposed to the air. This explains some of our observations not presented here, where we found that leaf temperatures often *increase* with increasing wind speed in the absence of shortwave radiation (darkness), while for a wet soil surface, increasing wind is usually associated with increasing evaporative cooling and hence decreasing surface temperature. For a leaf, the energy for transpiration in darkness is mainly supplied by sensible heat flux (on both sides), which increases with increasing wind speed. In contrast, the

energy for evaporation from a soil surface in darkness is supplied by sensible heat on the evaporating surface only, and by soil heat flux from below. In this case, increasing wind speed by itself does not lead to as much additional heat input for evaporation, until the surface cools, increasing the temperature gradient driving upwards soil heat flux. It may also be noteworthy in this context, that the expression for aerodynamic resistance ($r_a$) given by Monteith (1965, Eq. 14) has been pointed out by other authors to result in heat transfer 2.5 times higher than expected if interpreted as a one-sided resistance (Parlange et al., 1971).

This may have arisen from the confusion about one-sided vs. two-sided energy exchange. Our experimental results clearly illustrate that the inconsistencies we found in the PM and MU equations are not just semantic, but actually lead to very significant biases in simulated transpiration rates for known stomatal resistance, which would alternatively lead to biases in deduced resistance for known transpiration rates. The results further illustrate that our correction for two-sided leaves improves reproduction of leaf-scale measurements tremendously (MUc vs. PM in Fig. 7), but additional consideration of the surface

temperature-longwave emission feedback (Eq. 27 and Rlin in Fig. 7) is almost equally important to accurately capture the characteristics of the leaf energy balance.

  Although the up-scaling of a physically-based leaf-scale model to a canopy or land surface is fraught with various challenges, including characterisation of the stomatal or canopy conductance, canopy-scale boundary layer conductance, consideration of canopy storage and distinction between radiative and aerodynamic surface temperatures (Monteith, 1965; Tanner and Fuchs,

1968; Jarvis and McNaughton, 1986; Raupach, 1995; Mallick et al., 2013), we believe that care must be taken to start off with the correct leaf-scale model. In the present study, we have developed an experimental setup allowing to control all relevant boundary conditions at the leaf scale, including stomatal conductance, and measuring, to our knowledge for the first time, all components of the leaf energy balance. In contrast to previous tests of the PM-equation, which were conducted at the canopy scale, where boundary layer and canopy conductances could not be measured directly, we have been able to eliminate any need

for model calibration, and in this way discovered that the PM-equation, in its original formulation and common use, does not accurately represent leaf-scale processes. Our newly derived analytical solutions (Eqs. 27 and 22) not only more accurately reproduce leaf-scale sensible and latent heat fluxes, but they also allow direct calculation of leaf temperature, which could be used as an additional diagnostic variable at the canopy scale and also be further developed to improve remote-sensing based evaporation products.

Given the widespread and successful use of the PM-equation, the question arises whether common practice, which relies on parameterisation by fitting resistance terms that provide match with observations, somehow compensates for the errors we identified in the present study. The answer is "yes and no". As shown in Fig. 8, there are certain conditions, for which the PM-equation and the corrected solutions yield very similar results and one could easily obtain much closer match to the experimental results by fitting significantly larger values for $r_a$ and $r_s$ in the PM-equation than those estimated from diffusive



resistance of perforated surfaces (the laser perforations in our artificial leaves). However, the sensitivity of latent and sensible heat flux to changing atmospheric conditions (e.g. shortwave irradiance and air temperature) deduced form the PM-equation would clearly be different than the trends produced by the corrected equations and numerical simulations (Fig. 8). This suggests that use of the PM-equation for projections under future climate change scenarios could lead to a bias in the results. This potential source of bias could be reduced using the corrected equation presented in this study (irrespective of the estimated

resistance values fitted for a canopy).

## 6 Conclusions

In this study, we revisit the governing equations for the exchange of water vapour and energy between a planar leaf and a surrounding air stream under forced convection. We derived general analytical solutions for steady-state sensible and latent heat fluxes from a leaf and the corresponding leaf temperature (Eqs. 10–12, based on the approach by Penman (1952). The general

equation permits comparison between different analytical solutions available in the literature, by substituting appropriate formulations of the sensible and latent heat transfer coefficients. Our analysis reveals that the Penman-Monteith equation (Eq. 19), even with its modification by Monteith and Unsworth (2013) (Eq. 21) is not accurate for a typical planar leaf, due to omission of the radiative and sensible heat fluxes from one side of a leaf. We demonstrated how our general solution can be used to obtain a more consistent representation of leaf energy and gas exchange, in agreement with leaf-scale experimental data (using

artificial leaves). We propose that the same approach could prove useful to derive a more accurate canopy-scale representation of latent and sensible heat fluxes, considering their coupling with radiative exchange and ground heat flux. The new generalised leaf-scale equations offer a promise for more consistent responses of latent and sensible heat fluxes to changes in atmospheric forcing in future climates than the responses predicted by the original PM equation (due to the omissions therein).

## Appendix A: Tables of symbols

Tables A1 and A2 list all symbols used in this study, their descriptions, standard values and units.





**Table A1.** Table of greek symbols and standard values used in this paper. All area-related variables are expressed per unit leaf area.

| Variable | Description (value) | Units |
|---|---|---|
| $\alpha_a$ | Thermal diffusivity of dry air | $\frac{m^2}{s}$ |
| $\beta_B$ | Bowen ratio (sensible/latent heat flux) | 1 |
| $\gamma_v$ | Psychrometric constant | $\frac{Pa}{K}$ |
| $\Delta_{eTa}$ | Slope of saturation vapour pressure at air temperature | $\frac{Pa}{K}$ |
| $\epsilon$ | Water to air molecular weight ratio (0.622) | 1 |
| $\epsilon_l$ | Longwave emmissivity of the leaf surface (1.0) | 1 |
| $\lambda_E$ | Latent heat of evaporation (2.45e6) | $\frac{J}{kg}$ |
| $\nu_a$ | Kinematic viscosity of dry air | $\frac{m^2}{s}$ |
| $\rho_a$ | Density of dry air | $\frac{kg}{m^3}$ |
| $\rho_{al}$ | Density of air at the leaf surface | $\frac{kg}{m^3}$ |
| $\sigma$ | Stefan-Boltzmann constant (5.67e-8) | $\frac{J}{K^4 m^2 s}$ |



**Table A2.** Table of latin symbols and standard values used in this paper. All area-related variables are expressed per unit leaf area.

| Variable | Description (value) | Units |
|---|---|---|
| $a_s$ | Fraction of one-sided leaf area covered by stomata (1) | 1 |
| $a_{sh}$ | Fraction of projected area exchanging sensible heat with the air (2) | 1 |
| $c_E$ | Latent heat transfer coefficient | $\frac{J}{Pa\,m^2\,s}$ |
| $c_H$ | Sensible heat transfer coefficient | $\frac{J}{K\,m^2\,s}$ |
| $c_{pa}$ | Specific heat of dry air (1010) | $\frac{J}{K\,kg}$ |
| $C_{wa}$ | Concentration of water in the free air | $\frac{mol}{m^3}$ |
| $C_{wl}$ | Concentration of water in the leaf air space | $\frac{mol}{m^3}$ |
| $D_{va}$ | Binary diffusion coefficient of water vapour in air | $\frac{m^2}{s}$ |
| $E_l$ | Latent heat flux from leaf | $\frac{J}{m^2\,s}$ |
| $E_{l,mol}$ | Transpiration rate in molar units | $\frac{mol}{m^2\,s}$ |
| $E_w$ | Latent heat flux from a wet surface | $\frac{J}{m^2\,s}$ |
| $f_u$ | Wind function in Penman approach, f(u) adapted to energetic units | $\frac{J}{Pa\,m^2\,s}$ |
| $g$ | Gravitational acceleration (9.81) | $\frac{m}{s^2}$ |
| $g_{bw}$ | Boundary layer conductance to water vapour | $\frac{m}{s}$ |
| $g_{bw,mol}$ | Boundary layer conductance to water vapour | $\frac{mol}{m^2\,s}$ |
| $g_{sw}$ | Stomatal conductance to water vapour | $\frac{m}{s}$ |
| $g_{w,mol}$ | Stomatal conductance to water vapour | $\frac{mol}{m^2\,s}$ |
| $g_{tw}$ | Total leaf conductance to water vapour | $\frac{m}{s}$ |
| $g_{tw,mol}$ | Total leaf layer conductance to water vapour | $\frac{mol}{m^2\,s}$ |
| $N_{Gr_L}$ | Grashof number | 1 |
| $h_c$ | Average 1-sided convective transfer coefficient | $\frac{J}{K\,m^2\,s}$ |
| $H_l$ | Sensible heat flux from leaf | $\frac{J}{m^2\,s}$ |
| $k_a$ | Thermal conductivity of dry air | $\frac{J}{K\,m\,s}$ |
| $L_l$ | Characteristic length scale for convection (size of leaf) | $m$ |
| $N_{Le}$ | Lewis number | 1 |
| $M_{N_2}$ | Molar mass of nitrogen (0.028) | $\frac{kg}{mol}$ |
| $M_{O_2}$ | Molar mass of oxygen (0.032) | $\frac{kg}{mol}$ |
| $M_w$ | Molar mass of water (0.018) | $\frac{kg}{mol}$ |
| $n_{MU}$ | n=2 for hypostomatous, n=1 for amphistomatous leaves | 1 |
| $N_{Nu_L}$ | Nusselt number | 1 |
| $P_a$ | Air pressure | $Pa$ |





| Variable | Description (value) | Units |
|---|---|---|
| $P_{N2}$ | Partial pressure of nitrogen in the atmosphere | $Pa$ |
| $P_{O2}$ | Partial pressure of oxygen in the atmosphere | $Pa$ |
| $P_{wa}$ | Vapour pressure in the atmosphere | $Pa$ |
| $P_{was}$ | Saturation vapour pressure at air temperature | $Pa$ |
| $P_{wl}$ | Vapour pressure inside the leaf | $Pa$ |
| $N_{Pr}$ | Prandtl number (0.71) | 1 |
| $r_a$ | One-sided boundary layer resistance to heat transfer ($r_H$ in Monteith and Unsworth (2013, P. 231)) | $\frac{s}{m}$ |
| $r_{bw}$ | Boundary layer resistance to water vapour, inverse of $g_{bw}$ | $\frac{s}{m}$ |
| $R_{ll}$ | Longwave radiation away from leaf | $\frac{J}{m^2 s}$ |
| $R_{mol}$ | Molar gas constant (8.314472) | $\frac{J}{K mol}$ |
| $r_s$ | Stomatal resistance to water vapour (Monteith and Unsworth, 2013, P. 231) | $\frac{s}{m}$ |
| $R_s$ | Solar shortwave flux | $\frac{J}{m^2 s}$ |
| $r_{sw}$ | Stomatal resistance to water vapour, inverse of $g_{sw}$ | $\frac{s}{m}$ |
| $r_v$ | Leaf BL resistance to water vapour, (Monteith and Unsworth, 2013, Eq. 13.16) | $\frac{s}{m}$ |
| $N_{Re_L}$ | Reynolds number | 1 |
| $N_{Re_c}$ | Critical Reynolds number for the onset of turbulence (3000) | 1 |
| $S$ | Factor representing stomatal resistance in Penman (1952) | 1 |
| $N_{Sh_L}$ | Sherwood number | 1 |
| $T_a$ | Air temperature | $K$ |
| $T_l$ | Leaf temperature | $K$ |
| $T_w$ | Radiative temperature of objects surrounding the leaf | $K$ |
| $v_w$ | Wind velocity | $\frac{m}{s}$ |



## Appendix B: Mathematical derivations

**B1   Boundary layer conductance to water vapour**

The total leaf conductance to water vapour is determined by the boundary layer and stomatal conductances and equal to 1 over the sum of their respectife resistances ($g_{tw} = 1/(r_{sw} + r_{bw})$. The boundary layer conductance for water vapour is equivalent to the mass transfer coefficient for a wet surface (Incropera et al., 2006, Eq. 7.41):

$$g_{bw} = N_{Sh_L} D_{va}/L_l \tag{B1}$$

where $N_{Sh_L}$ is the dimensionless Sherwood number and $D_{va}$ is the diffusivity of water vapour in air. If the convection coefficient for heat is known, the one for mass ($g_{bw}$) can readily be calculated from the relation (Incropera et al., 2006, Eq. 6.60):

$$g_{bw} = \frac{a_s h_c}{\rho_a c_{pa} N_{Le}^{1-n}} \tag{B2}$$

where $a_s$ is the fraction of one-sided transpiring surface area in relation to the surface area for sensible heat exchange, $c_{pa}$ is the
constant-pressure heat capacity of air, $n$ is an empirical constant ($n = 1/3$ for general purposes) and $N_{Le}$ is the dimensionless Lewis number, defined as (Incropera et al., 2006, Eq. 6.57):

$$N_{Le} = \alpha_a/D_{va} \tag{B3}$$

where $\alpha_a$ is the thermal diffusivity of air. The value of $a_s$ was set to 1 for leaves with stomata on one side only, and to 2 for stomata on both sides. Other values could be used for leaves only partly covered by stomata.

**B2   Effect of leaf temperature on the leaf-air vapour concentration gradient**

The concentration difference in Eq. 5 is a function of the temperature and the vapour pressure differences between the leaf and the free air. Assuming that water vapour behaves like an ideal gas, we can express its concentration as:

$$C_{wl} = \frac{P_{wl}}{R_{mol}T_l} \tag{B4}$$

where $P_{wl}$ is the vapour pressure inside the leaf, $R_{mol}$ is the universal gas constant and $T_l$ is leaf temperature. A similar
relation holds for the vapour concentration in free air, $C_{wa} = P_{wa}/(R_{mol}T_l)$. In this study the vapour pressure inside the leaf is assumed to be the saturation vapour pressure at leaf temperature, which is computed using the Clausius-Clapeyron relation (Hartmann, 1994, Eq. B.3):

$$P_{wl} = 611 \exp\left( \frac{\lambda_E M_w}{R_{mol}} \left( \frac{1}{273} - \frac{1}{T_l} \right) \right) \tag{B5}$$

where $\lambda_E$ is the latent heat of vaporisation and $M_w$ is the molar mass of water.

Note that the dependence of the leaf-air water concentration difference ($C_{wl} - C_{wa}$) in Eq. B4 is very sensitive to leaf temperature. For example, if the leaf temperature increases by 5K relative to air temperature, $C_{wl} - C_{wa}$ would double, while if leaf temperature decreased by 6K, $C_{wl} - C_{wa}$ would go to 0 at 70% relative humidity (Fig. A1).





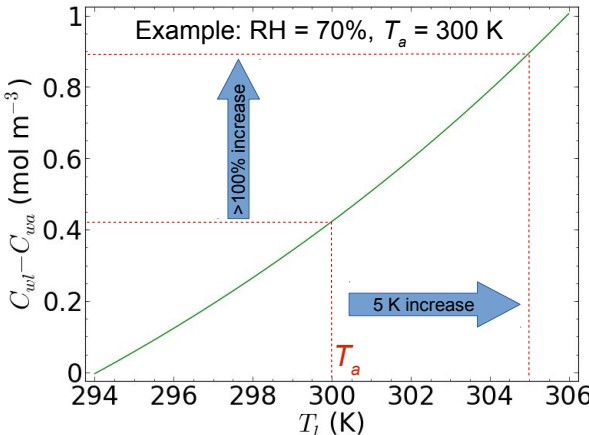

**Figure A1.** Dependence of the leaf-air water vapour concentration difference ($C_{wl} - C_{wa}$) on leaf temperature ($T_l$). In this example (70% relative humidity, 300 K air temperature ($T_a$), the water vapour concentration difference doubles for an increase in leaf temperature by 5 K relative to air temperature, or drops to 0 for a decrease in leaf temperature by 6 K. Plot obtained by inserting Eq. B5 into Eq. B4. $C_{wa}$ was obtained substituting $T_a$ for $T_l$ and multiplying Eq. B5 by the assumed relative humidity of 0.7.

## B3    Concentration or vapour pressure gradient driving transpiration?

Note that $E_{l,mol}$ is commonly expressed as a function of the vapour pressure difference between the free air ($P_{wa}$) and the leaf

($P_{wl}$), in which the conductance ($g_{tw,mol}$) is expressed in molar units (mol m$^{-2}$ s$^{-1}$):

$$E_{l,mol} = g_{tw,mol} \frac{P_{wl} - P_{wa}}{P_a} \tag{B6}$$

For $P_{wl} = P_{wa}$, Eq. 5 can still give a flux, whereas Eq. B6 gives zero flux. This is because the concentrations of vapour in air (mol m$^{-3}$) can differ due to differences in tempertaure, even if the partial vapour pressures are the same (see Eq. B4). Therefore, the relation between $g_{tw}$ and $g_{v,mol}$ has an asymptote at the equivalent temperature. It can be obtained by combining Eqs. 5

and B6 and solving for $g_{tw,mol}$:

$$g_{tw,mol} = g_{tw} \frac{P_a(P_{wa}T_l - P_{wl}T_a)}{(P_{wa} - P_{wl})R_{mol}T_aT_l} \tag{B7}$$

For $T_l = T_a$, the relation simplifies to:

$$g_{tw,mol} = g_{tw} \frac{P_a}{R_{mol}T_a} \tag{B8}$$

which, for typical values of $P_a$ and $T_a$ amounts to $g_{tw,mol} \approx 40$ mol m$^{-3}g_{tw}$. For all practical purposes, we found that Eqs. 5

and B6 with $g_{tw,mol} = g_{tw} \frac{P_a}{R_{mol}T_a}$ give similar results when plotted as functions of leaf temperature.





## B4 Model closure

Given climatic forcing as $P_a$, $T_a$, $R_s$, $P_{wa}$ and $v_w$, and leaf-specific parameters $a_s$, $a_{sH}$, $L_l$ and $g_{sw}$, we need to compute $C_{wa}$, $h_c$, $g_{bw}$ and a series of other derived variables, as described below.

The vapour concentration in the free air can be computed from vapour pressure analogously to Eq. B4:

$$C_{wa} = \frac{P_{wa}}{R_{mol} T_a} \qquad \text{(B9)}$$

The heat transfer coefficient ($h_c$) for a flat plate can be determined using the non-dimensional Nusselt number ($N_{Nu_L}$):

$$h_c = k_a \frac{N_{Nu_L}}{L_l} \qquad \text{(B10)}$$

where $k_a$ is the thermal conductivity of the air in the boundary layer and $L_l$ is a characteristic length scale of the leaf.

For sufficiently high wind speeds, inertial forces drive the convective heat transport (forced convection) and the relevant di-

400 mensionless number is the Reynolds number ($N_{Re_L}$), which defines the balance between inertial and viscous forces (Incropera et al., 2006, Eq. 6.41):

$$N_{Re_L} = \frac{v_w L_l}{\nu_a} \qquad \text{(B11)}$$

where $v_w$ is the wind velocity (m s$^{-1}$), $\nu_a$ is the kinematic viscosity of the air and $L_l$ is taken as the length of the leaf in wind direction.

In the absence of wind, buoyancy forces, driven by the density gradient between the air at the surface of the leaf and the free air dominate convective heat exchange ("free" or "natural convection"). The relevant dimensionless number here is the Grashof number ($N_{Gr_L}$), which defines the balance between buoyancy and viscous forces (Incropera et al., 2006, Eqs. 9.3 and 9.65):

$$N_{Gr_L} = \frac{g\left(\frac{\rho_a - \rho_{al}}{\rho_{al}}\right) L_l^3}{\nu_a^2} \qquad \text{(B12)}$$

where $g$ is gravity, while $\rho_a$ and $\rho_{al}$ are the densities of the gas in the atmosphere and at the leaf surface respectively.

For $N_{Gr_L} \ll N_{Re_L}^2$, forced convection is dominant and free convection can be neglected, whereas for $N_{Gr_L} \gg N_{Re_L}^2$ free convection is dominant and forced convection can be neglected (Incropera et al., 2006, P. 565). For simplicity, the analysis is limited to forced conditions, which is satisfied by considering wind speeds greater than 0.5 m s$^{-1}$ for $5 \times 5$ cm leaves.

The average Nusselt number under forced convection was calculated as a function of the average Reynolds number and a

415 critical Reynolds number ($N_{Re_c}$) that determines the onset of turbulence and depends on the level of turbulence in the free air stream or leaf surface properties (Incropera et al., 2006, P. 412)

$$N_{Nu_L} = (0.037 N_{Re_L}^{4/5} - C_1) N_{Pr}^{1/3} \qquad \text{(B13)}$$

with

$$C_1 = 0.037 C_2^{4/5} - 0.664 C_2^{1/2} \qquad \text{(B14)}$$





and

$$C_2 = \frac{N_{Re_L} + N_{Re_c} - |N_{Re_c} - N_{Re_L}|}{2} \tag{B15}$$

Eq. B15 was introduced to make Eq. B13 valid for all Reynolds numbers, and following considerations explained in our previous work (Schymanski et al., 2013), we chose $N_{Re_c} = 3000$ in the present simulations.

In order to simulate steady state leaf temperatures and the leaf energy balance terms using the above equations, it is necessary to calculate $\rho_a$, $D_{va}$, $\alpha_a$, $k_a$, and $\nu_a$, while $L_l$, $Re_c$ and $g_{sv}$ are input parameters, and $P_{wa}$ and $v_w$ (vapour pressure and wind speed) are part of the environmental forcing. $D_{va}$, $\alpha_a$, $k_a$ and $\nu_a$ were parameterised as functions of air temperature ($T_a$) only, by fitting linear curves to published data (Monteith and Unsworth, 2007, Table A.3):

$$D_{va} = (1.49 \times 10^{-7})T_a - 1.96 \times 10^{-5} \tag{B16}$$

$$\alpha_a = (1.32 \times 10^{-7})T_a - 1.73 \times 10^{-5} \tag{B17}$$

$$k_a = (6.84 \times 10^{-5})T_a + 5.62 \times 10^{-3} \tag{B18}$$

$$\nu_a = (9 \times 10^{-8})T_a - 1.13 \times 10^{-5} \tag{B19}$$

Assuming that air and water vapour behave like an ideal gas, and that dry air is composed of 79% $N_2$ and 21% $O_2$, we calculated air density as a function of temperature, vapour pressure and the partial pressures of the other two components using the ideal gas law:

$$\rho_a = \frac{n_a M_a}{V_a} = M_a \frac{P_a}{R_{mol} T_a} \tag{B20}$$

where $n_a$ is the amount of matter (mol), $M_a$ is the molar mass (kg mol$^{-1}$), $P_a$ the pressure, $T_a$ the temperature and $R_{mol}$ the molar universal gas constant. This equation was used for each component, i.e. water vapour, $N_2$ and $O_2$, where the partial pressures of $N_2$ and $O_2$ are calculated from atmospheric pressure minus vapour pressure, yielding:

$$\rho_a = \frac{M_w P_{wa} + M_{N_2} P_{N_2} + M_{O_2} P_{O_2}}{R_{mol} T_a} \tag{B21}$$

where $M_{N_2}$ and $M_{O_2}$ are the molar masses of nitrogen and oxygen respectively, while $P_{N_2}$ and $P_{O_2}$ are their partial pressures, calculated as:

$$P_{N_2} = 0.79(P_a - P_{wa}) \tag{B22}$$

and

$$P_{O_2} = 0.21(P_a - P_{wa}) \tag{B23}$$



## B5 Analytical solutions by Penman

In order to obtain analytical expressions for the different leaf energy balance components, one would need to solve the leaf energy balance equation for leaf temperature first. However, due to the non-linearities of the blackbody radiation and the saturation vapour pressure equations, an analytical solution has not been found yet. Penman (1948) proposed a work-around, which we reproduced below, adapted to our notation and to a wet leaf, while Penman's formulations referred to a wet soil surface. He formulated evaporation from a wet surface as a diffusive process driven by the vapour pressure difference near the wet surface and in the free air:

$$E_w = f_u(P_{wl} - P_{wa}) \tag{B24}$$

where $E_w$ (J s$^{-1}$ m$^{-2}$) is the latent heat flux from a wet surface and $f_u$ is commonly referred to as the wind function. Penman then defined the Bowen ratio as (Eq. 10 in Penman (1948)):

$$\beta_B = H_l/E_w = \gamma_v \frac{T_l - T_a}{P_{wl} - P_{wa}} \tag{B25}$$

where $H_l$ is the sensible heat flux and $\gamma_v$ is the psychrometric constant, referring to the ratio between the transfer coefficients for sensible heat and that for water vapour.

In order to eliminate $T_l$, Penman introduced a term for the ratio of the vapour pressure difference between the surface and the saturation vapour pressure at air temperature ($P_{was}$) to the temperature difference between the surface and the air:

$$\Delta_{eTa} = \frac{P_{wl} - P_{was}}{T_l - T_a} \tag{B26}$$

and he proposed to approximate this term by the slope of the saturation vapour pressure curve evaluated at air temperature, which can be obtained by substitution of $T_a$ for $T_l$ and differentiation of Eq. B5 with respect to $T_a$:

$$\Delta_{eTa} = \frac{611\lambda_E M_w \exp\left(\frac{\lambda_E M_w}{R_{mol}}\left(\frac{1}{273} - \frac{1}{T_a}\right)\right)}{R_{mol} T_a^2} \tag{B27}$$

For further discussion of the meaning of this assumption, please refer to Mallick et al. (2014).

Susbstitution of Eq. 9 in Eq. B25 yields (Eq. 15 in Bowen (1926)):

$$\beta_B = \frac{\gamma_v}{\Delta_{eTa}} \frac{(P_{wl} - P_{was})}{(P_{wl} - P_{wa})} \tag{B28}$$

Substituting $E_w$ for $E_l$ in the energy balance equation (Eq. 1), inserting $H_l = \beta_B E_w$ (Eq. B25) and solving for $E_w$ gives:

$$E_w = \frac{R_s - R_{ll}}{\beta_B + 1} \tag{B29}$$

Substitution of Eq. B28 into Eq. B29, equating with Eq. B24 and solving for $P_{wl}$ gives:

$$P_{wl} = \frac{f_u(\Delta_{eTa} P_{wa} + \gamma_v P_{was}) + \Delta_{eTa}(R_s - R_{ll})}{f_u(\Delta_{eTa} + \gamma_v)} \tag{B30}$$





Now, insertion of Eq. B30 into Eq. B24 gives the so-called "Penman equation" :

$$E_w = \frac{\Delta_{eTa}(R_s - R_{ll}) + f_u \gamma_v (P_{was} - P_{wa})}{\Delta_{eTa} + \gamma_v} \tag{B31}$$

Eq. 15 is equivalent to Eq 16 in Penman (1948), but Eq. 17 in Penman (1948), which should be equivalent to Eq. B30, has $P_{wl}$ ($e_s$ in Penman's notation) on both sides, so it seems to contain an error. In his derivations, Penman expressed $R_s - R_{ll}$ as "net radiant energy available at surface" and pointed out that the above two equations can be used to estimate $E_l$ and $T_l$ from air conditions only. This neglects the fact that $R_{ll}$ is also a function of the leaf temperature. To estimate surface temperature, Eq. B30 can be inserted into Eq. 9 and solved for $T_l$, yielding:

$$T_l = \frac{R_s - R_{ll} + f_u(\gamma_v T_a + \Delta_{eTa} T_a + P_{wa} - P_{was})}{f_u(\gamma_v + \Delta_{eTa})} \tag{B32}$$

### B5.1 Introduction of stomatal resistance by Penman (1952)

To account for stomatal resistance to vapour diffusion, Penman (1952) introduced an additional multiplicator ($S$) in Eq. B24 (Penman, 1952, Appendix 13):

$$E_l = f_u S(P_{wl} - P_{wa}) \tag{B33}$$

where $S = 1$ for a wet surface (leading to Eq. B24) and $S < 1$ in the presence of significant stomatal resistance.

In accordance with Eqs. B24 and B25, $H_l$ can be expressed as (Penman, 1952, Appendix 13):

$$H_l = \gamma_v f_u(T_l - T_a) \tag{B34}$$

Substitution of Penman's simplifying assumption ($T_l - T_a = (P_{wl} - P_{was})/\Delta_{eT}$, Eq. 9) is the first step to eliminating $T_l$:

$$H_l = \frac{\gamma_v f_u(P_{wl} - P_{was})}{\Delta_{eTa}} \tag{B35}$$

A series of algebraic manipulations involving Eqs. B33, B35 and 1 and the resulting Eq. B36 is given in Penman (1952, Appendix 13). When solving Eqs. B33, B35 and 1 for $E_l$, $H_l$ and $P_{wl}$, we obtained:

$$E_l = \frac{S\Delta_{eTa}(R_s - R_{ll}) + S\gamma_v f_u(P_{was} - P_{wa})}{S\Delta_{eT} + \gamma_v} \tag{B36}$$

$$H_l = \frac{\gamma_v(R_s - R_{ll}) + S\gamma_v f_u(P_{wa} - P_{was})}{S\Delta_{eTa} + \gamma_v} \tag{B37}$$

$$P_{wl} = \frac{(\Delta_{eTa}/f_u)(R_s - R_{ll}) + (S\Delta_{eTa} P_{wa} + \gamma_v P_{was})}{S\Delta_{eTa} + \gamma_v} \tag{B38}$$





## B5.2    Analytical solutions for leaf temperature, $f_u$, $\gamma_v$ and $S$

Equation B38 can be inserted into Eq. 9 and solved for leaf temperature to yield:

$$T_l = T_a + \frac{R_s - R_{ll} - S f_u (P_{was} - P_{wa})}{f_u (S \Delta_{eT} + \gamma_v)} \tag{B39}$$

Penman (1952) proposed to obtain values of $f_u$ and $S$ for a plant canopy empirically and described ways how to do this.
However, for a single leaf, $f_u$ and $S$ could also be obtained analytically from our detailed mass and heat transfer model.

Comparison of Eq. B33 with Eq. B6 (after substituting Eq. 4) reveals that $S$ is equivalent to:

$$S = \frac{M_w g_{tw,mol} \lambda_E}{P_a f_u} \tag{B40}$$

where $f_u$ was defined by Penman (1948) as the transfer coeffient for wet surface evaporation, i.e. a function of the boundary layer conductance only.

To find a solution for $f_u$, we first formulate $E_w$ as transpiration from a leaf where $g_{tw} = g_{bw}$, using Eqs. 4, B6 and B8:

$$E_w = \frac{\lambda_E M_w g_{bw}}{R_{mol} T_a}(P_{wl} - P_{wa}) \tag{B41}$$

Comparison of Eq. B41 with B24 gives $f_u$ as a function of $g_{bw}$:

$$f_u = g_{bw} \frac{\lambda_E M_w}{R_{mol} T_a} \tag{B42}$$

Comparison of Eq. B34 and Eq. 3 reveals that

$$\gamma_v = \frac{a_{sh} h_c}{f_u}, \tag{B43}$$

and insertion of Eqs. B42 and B2 give $\gamma_v$ as a function of $a_{sh}$ and $a_s$:

$$\gamma_v = a_{sh}/a_s \frac{N_{Le}^{\frac{2}{3}} R_{mol} T_a \rho_a c_{pa}}{\lambda_E M_w} \tag{B44}$$

Now, we can insert Eqs. B42, B8 and 6 into Eq. B40 to obtain $S$ as a function of $g_{sw}$ and $g_{bw}$:

$$S = \frac{g_{sw}}{g_{bw} + g_{sw}} \tag{B45}$$

The above equation illustrates that $S$ is not just a function of stomatal conductance, but also the leaf boundary layer conductance, explaining why Penman (1952) found that $S$ depends on wind speed.

## B6    Psychrometric constant in the Penman-Monteith equation

Monteith and Unsworth (2013) provide a definition of $\gamma_v$ as:

$$\gamma_v = \frac{c_{pa} P_a}{\lambda_E \epsilon} \tag{B46}$$





where $\epsilon$ is the ratio of molecular weights of water vapour and air (given by Monteith and Unsworth (2013) as 0.622). The molar mass of air is $M_a = \rho_a V_a / n_a$, while according to the ideal gas law, $V_a/n_a = R_{mol} T_a / P_a$, which yields for $\epsilon = M_w / M_a$:

$$\epsilon = \frac{M_w P_a}{R_{mol} T_a \rho_a} \tag{B47}$$

Inserting Eqs. B21, B22 and B23 into the above, $T_a$ cancels out, and at standard atmospheric pressure of 101325 Pa, we obtain values for $\epsilon$ between 0.624 and 0.631 for vapour pressure ranging from 0 to 3000 Pa, compared to the value of 0.622 mentioned by Monteith and Unsworth (2013).

### B7 Meaning of resistances in PM equation

As opposed to the formulations in Section 2.1, where sensible and latent heat transfer coefficients ($h_c$ and $g_{tw}$ respectively) translate leaf-air differences in temperature or vapour concentration to fluxes, resistances in the PM equation are defined in the context of the following two equations (Monteith and Unsworth, 2013, Eqs. 13.16 and 13.20):

$$E_l = \frac{a_s \lambda_E \rho_a \epsilon}{P_a (r_v + r_s)} (P_{wl} - P_{wa}) \tag{B48}$$

and

$$H_l = \frac{a_{sh} \rho_a c_{pa}}{r_a} (T_l - T_a), \tag{B49}$$

where $r_v$ and $r_s$ are the one-sided leaf boundary layer and stomatal resistances to water vapour respectively, and $r_a$ is the one-sided leaf boundary layer resistance to sensible heat transfer. Note that we introduced $a_s$, $a_{sh}$ and $r_s$ in Eqs. B48 and B49 based on the description on P. 231 in Monteith and Unsworth (2013), where the authors also assumed that $r_v \approx r_a$.[1]

Comparison of Eq. B48 (after substitution of Eq. B47) with our fundamental diffusion equation (Eq. 5, after substitution of Eqs. B4 and B9 and insertion into Eq. 4) reveals that under isothermal conditions ($T_l = T_a$):

$$r_v = a_s / g_{bw}, \tag{B50}$$

while comparison of Eq. B49 with Eq. 3 reveals that

$$r_a = \frac{\rho_a c_{pa}}{h_c}. \tag{B51}$$

### B8 Comparison of our general analytical solution with original Penman and Penman-Monteith equations

From the general form (Eq. 10), we can recover most of the above analytical solutions by appropriate substitutions for $c_E$ and $c_H$, but closer inspection of the necessary substitutions reveals some inconsistencies.

The Penman equation for a wet surface (Eq. 15) can be recovered by substituting $c_E = f_u$ and $c_H = \gamma_v f_u$ into Eq. 10 (Fig. 3a), while additional substitution of Eq. 17 leads to recovery of Eq. 16, the Penman equation, as reformulated by Monteith (1965). The formulation for leaf transpiration derived by Penman (1952) (Eq. B36) is obtained by substituting $c_E = S f_u$

---

[1]Division of Eq. B51 by Eq. B50 and substitution of Eqs. B2, B3, B17 and B16 reveals that $r_a / r_v = N_{Le}^{-2/3} = 1.082$.





(deduced from Eq. B33) and $c_H = \gamma_v f_u$ (from Eq. B34). These substitutions are consistent with the formulations of latent and sensible heat flux given in Eqs. B34 and B24 or B33, as long as $f_u$ and $r_a$ refer to the *total resistances* of a leaf to latent and sensible heat flux respectively, as Eq. 17 in conjunction with $c_H = \gamma_v f_u$ implies that:

$$c_H = (\rho_a c_{pa})/r_a \qquad\qquad\qquad (B52)$$

Similarly, the Penman-Monteith equation (Eq. 19 with $\gamma_v$ defined in Eq. B46) could be recovered by substituting $c_E = \epsilon \lambda_E \rho_a/(P_a(r_s + r_v))$ and $c_H = c_{pa}\rho_a/r_a$, with subsequent substitution of $r_v = r_a$. Note however, that these substitutions are not consistent with Eqs. B48 and B49, as the factors $a_s$ and $a_{sh}$ (referring to the number of leaf faces exchanging latent and sensible heat flux respectively) are missing (Fig 3b cf. 3c). This is because the PM equation was derived with a soil surface in mind, which exchanges latent and sensible heat only on one side, and hence is not appropriate for a leaf. To alleviate this constraint, one could define $r_a$ and $r_s$ as total (two-sided) leaf resistances, but in this case, the simplification $r_v \approx r_a$ is not valid for hypostomatous leaves, as $r_a$ would then be only half of $r_v$. This is illustrated in Fig. 3c, where sensible heat flux is released from both sides of the leaf, while latent heat flux is only released from the abaxial side, implying that $a_{sh} = 2$ and $a_s = 1$.

Monteith and Unsworth (2013) acknowledged that a hypostomatous leaf could exchange sensible heat on two sides, but latent heat on one side only and introduced the parameter $n_{MU} = a_{sh}/a_s$ to account for this (Eq. 21). Using our general equation, it should be possible to reproduce the MU-Equation (Eq. 21) by substituting $c_E = a_s \epsilon \lambda_E \rho_a/(P_a(r_s + r_v))$ (deduced from Eq. B48) and $c_H = a_{sh} c_{pa}\rho_a/r_a$ (deduced from Eq. B49) into Eq. 10. However, the result of this substitution, as presented in Eq. B53, is not the same as Eq. 21 after substitution of Eqs. B46 and $n_{MU} = a_{sh}/a_s$, which would result in Eq. B54:

$$E_l = \frac{a_s \epsilon \lambda_E \big(\Delta_{eTa} r_a (R_s - R_{ll}) + a_{sh} c_{pa}\rho_a (P_{was} - P_{wa})\big)}{P_a a_{sh} c_{pa}(r_s + r_a) + \Delta_{eTa} a_s \epsilon \lambda_E} \qquad (B53)$$

vs.

$$E_l = \frac{a_s \epsilon \lambda_E \big(\Delta_{eTa} r_a (R_s - R_{ll}) + c_{pa}\rho_a (P_{was} - P_{wa})\big)}{P_a a_{sh} c_{pa}(r_s + r_a) + \Delta_{eTa} a_s \epsilon \lambda_E} \qquad (B54)$$

Note the missing $a_{sh}$ in the nominator of Eq. B54, as pointed out in the main text.

## B9 Surface temperature-dependence of net radiation

In the main text, Eq. 2 was linearised by taking its derivative with respect to $T_l$, defining this derivative as the slope of a linear function of temperature with an intercept chosen to make this function intersect with Eq. 2 at $T_l = T_a$. The result is given in Eq. 23 and plotted in Fig. A3.

## B10 Calculation of stomatal conductance from pore dimensions

At least three confocal laser scanning images of each perforated foil were analysed and average pore area ($A_p$, m$^2$), pore
radius ($r_p$, m), number of pores per surface area ($n_p$, m$^{-2}$) and average distance to nearest neighbour ($s_p$, m) was computed





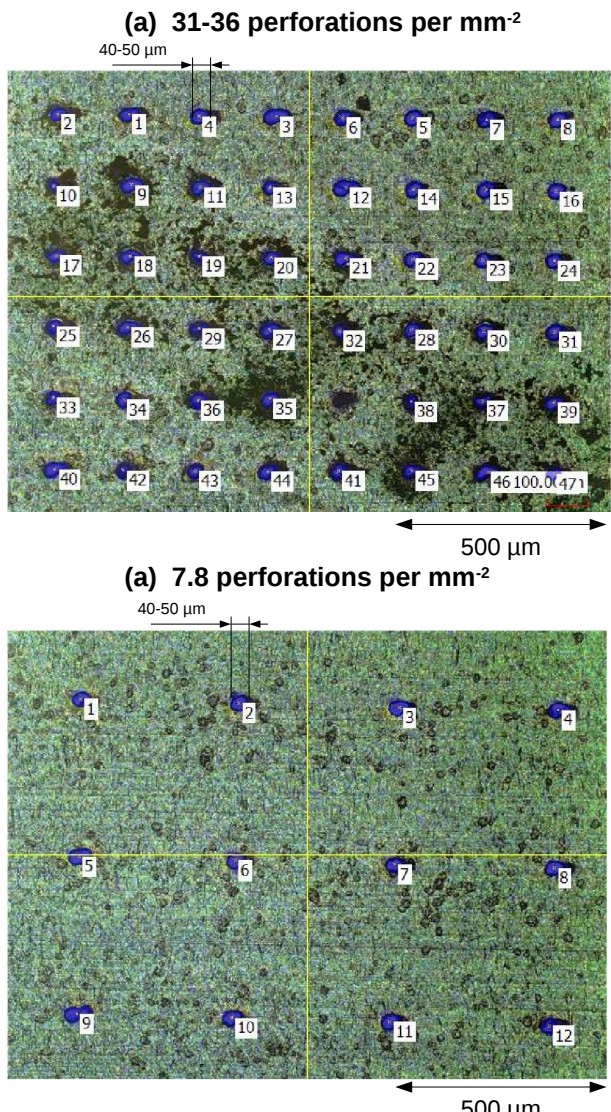

**Figure A2.** Confocal laser scanning microscope (CLSM) images of perforated foils summarised in Tab. 1. Blue coloured and numbered patches represent the identified perforations in each picture.

for each image. The resulting stomatal conductance was computed following the derivations presented by Lehmann and Or (2015), assuming that the stomatal conductance results from two resistances in series, the throat resistance ($r_{sp}$), dependent on the areas of the pores and the thickness of the perforated foil ($d_p$), and the vapour shell resistance ($r_{vs}$), dependent on the size and spacing of the stomata, which can be understood as the resistance related to distribution of the point source water vapour over the entire one-sided leaf boundary layer. We hereby neglect any internal resistance (termed "end correction" by Lehmann





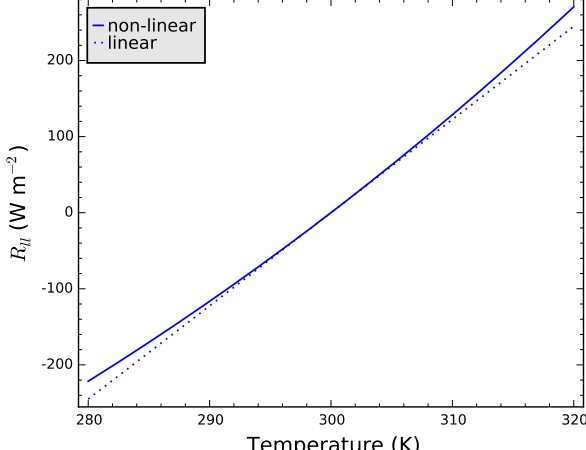

**Figure A3.** Net longwave radiation away from leaf as a function of leaf temperature. Solid line represents Eq. 2, while the dotted line represents the linearised function (Eq. 23). Calculations are based on 300 K air and wall temperature ($T_a$ and $T_w$ respectively).

and Or (2015)), as we assume that the wet filter paper has direct contact with the perforated foil. The throat resistance was computed as (Eq. 1 in Lehmann and Or, 2015):

$$r_{sp} = \frac{d_p}{A_p k_{dv} n_p} \tag{B55}$$

where $k_{dv}$ is the ratio of the vapour diffusion coefficient and the molar volume of air ($D_{va}/V_m$), and $A_p = \pi r_p^2$. For the vapour
shell resistance, we use the formulation originally proposed by Bange (1953):

$$r_{vs} = \left( \frac{1}{4 r_p} - \frac{1}{\pi s_p} \right) \frac{1}{k_{dv} n_p} \tag{B56}$$

where $s_p$ (m) is the spacing between stomata, inferred from the images as $s_p = 1/\sqrt{n_p}$. Stomatal conductance ($g_{sw}$) for each image was then calculated following Eq. B8, i.e. $g_{sw} = g_{sw,mol} R_{mol} T_a / P_a$, after substituting $g_{sw,mol} = 1/(r_{sp} + r_{vs})$.

*Author contributions.* SJS performed the mathematical derivations, designed and carried out the experiments and wrote the paper. DO was
590 involved in the design of the experimental setup, interpretation of the results and writing the paper.

*Acknowledgements.* The authors are very grateful to Dani Breitenstein for his assistance in designing and constructing the wind tunnel, to Stefan Meier and Joni Dehaspe for assistance in constructing artificial leaves, and to Hans Wunderli for assistance in the lab. We also wish to acknowledge technical advice by Roland Kuenzli (DMP AG, Fehraltorf, Switzerland), laser perforation services by Ralph Beglinger (Lasergraph AG, Würenlingen, Switzerland), Robert Voss (ETH Zurich, Switzerland) and Rolf Brönnimann (EMPA, Zurich, Switzerland)
as well as funding by the Swiss National Science Foundation, Project 200021 135077.



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
