# Peer review of "Leaf-scale experiments reveal important omission in the Penman-Monteith equation"

_Hydrology and Earth System Sciences, 2016_

## Short Comment (SC1) · 10 Aug 2016

It's great to see a back-to-basics assessment of the water and heat fluxes of a leaf and I commend you for that. I am sure it is a good time to go back to studying the aerodynamic resistances of sensible and latent heat fluxes. My only substantial comment is this: if you are going to go into the physics and mathematics of whether there are stomata on both sides of the leaf or not, should not you also consider the impact of the fact that the leaves usually are held perpendicular to the sun's rays? This means that one side of the leaf is sunlit and the other is shaded giving substantially different surface temperatures. Does this not affect the equations too? How was the sun-shine included in the wind tunnel? Would it make a difference if the short wave radiation is

diffuse?

---

## Author Comment (AC1) · 23 Aug 2016

Thank you for the supportive comment. You have made a very good point, which applies not only to irradiated leaves, but even to our experiments in darkness, using leaves with stomata on one side only. Due to evaporative cooling on one side, there can indeed be a temperature gradient between the two sides of a leaf, which is strongly dependent on leaf thickness and its thermal conductance. We note that both the numerical model used in this study and the analytical solutions based on Penman and Monteith neglect this gradient and assume that both sides of the leaf are at the same temperature. For natural leaves of 0.1-0.5 mm in thickness that are fully hydrated, the temperature gradient would not exceed 0.1- 0.5 K, which justifies the assumption of single temperature for both sides. We have added a conductive heat flux across the leaf to the numerical solution (in a separate study) and found it had only a minor effect on the net exchange of sensible and latent heat fluxes. Even for artificial leaves (perforated aluminum foil with filter paper sandwiched in the middle) with potential air gaps and lower thermal conductance than real leaves, the measured "leaf temperature" was only 1K different between the two sides of the leaf surfaces. Moreover, the simulated leaf temperatures considering heat transport between the two surfaces bracket the temperature when the heat exchange is neglected. In other words, neglecting conductive heat flux across a thin, planar hydrated leaf is justified for representing steady-state latent and sensible heat exchange with the atmosphere.

Nevertheless, we will explain this point in more detail in a follow-up technical paper on the wind-tunnel and artificial leaves, and also highlight the challenges in including irradiance in the wind tunnel experiments. Our preliminary measurements have shown that while the latent heat flux under irradiance was consistent between wind tunnel experiments and models, the sensible heat flux could not be measured correctly. This deviation was attributed to stray irradiance absorbed by the wind tunnel walls. We are working on solutions to reduce stray shortwave irradiance (by focusing the beam on the leaf surface only) and results will be presented in the technical note.

---

## Referee Comment (RC1) · S.C. Dekker (Referee) · 9 Sep 2016

Review Schymanski and Or Hess

A wonderful paper which definitely is worth to publish in HESS. It is great that the lab of Schymanski and Or have the possibility to do fundamental research: it is amazing to see how artificial made leaves with known conductance are tested in wind tunnels to finally address if one of the most used equations in hydrology is correct or not. I definitely support to publish this paper. In general, I think that the authors could show more results and make some extra interpretation on that in the discussion. This might overlap with the planned technical note, but it would strengthen this paper.

[Figure]

i) Abstract: L9: you report two errors in the PM, without telling which ones (leaf exchanges of sensible heat and longwave radiation). Please add them, a reader would like to know what is missing.

ii) Also in the abstract I would like to see a statement that the error can be enormous. For instance, in Fig 7,8 the authors report differences in modelled vs observed ET and H of up to 100 W/m2

iii) Minor point: Figure 1, you write the Eq Rs=El+Hl+Rll, while in the text you write Rs=Rll+Hl+El

iv) There are some inconsistencies in reporting units in the main text. I mean all units are correct but are not reported. I would be in favour that all variables if introduced for the first time should have a unit. For instance, no unit is given for Rs, while El (L71) does have a unit.

v) L. 80-84, This part is introduced to fast. I do not understand this sentence... you introduce hc here (convective transport coefficient) What do you mean with leaf boundary conductance (the leaf boundary layer conductance to sensible heat is gsw (ant not hc)

vi) I think it would be good to end chapter 2 with a new section (2.5) on the comparison of the different models (Rlin, MuC,PM, MU) and that you finally want to compare numerical solutions with PM and your new derived analytical solution (Rlin). This means that the part from the results L258-264, which are actually methods, should be incorporated in this new section (2.5). It will help the reader what he/she can expect.

vii) Make new section 4.1: I think that the results should be divided in two parts. First 4.1 in which the data are compared to numerical solutions. In that sense I also would like to incorporate the data wind speed against H and ET measurements and numerical solutions. In this section it would be good to link back to Figure 1, were they show that H, ET are dependent on wind speed and Rs and Rll not

viii) Make new section 4.2: In section 4.2 you then can compare the different models. If possible I would like to see 4 graphs (35 pores/mm2 against wind speed, 7 pores/mm2 against wind speed, 35 pores/mm2 against VPD, 7 pores/mm2 against VPD). If they are not measured then only the 35 and 7 against VPD and 35 against wind speed. In this section I hope that the authors can tell a little bit more on the general behaviour of the models and why there is a clear order of over/under estimation depending on the models used. The authors could do that by also showing the resulted Rll ( (Eq2) versus Eq. (23) and as a result also the difference in Tl for the different models. Then I hope it becomes more clear why the PM, MUc,MU over and/or underestimate.

ix) Minor point: Results (L257). Here we only report two experiments under varying vapour pressure. This is true for Figure 6, but not for Figure 7 as here you also report an effect of different wind speeds. Please add that.

x) Minor point: I am not in favour of saying that most of the results will be presented in a technical note

xi) L 288, again two errors without clearly telling which ones.

xii) L 292: The discussion directly continues in observations which are not shown, while I think that the authors could discuss the observations which are presented. Especially why some models over/under estimate the results is important to address and important for the readers of HESS

Appendix: It is a real pleasure that, as far I can see, all equations and units are correct. One minor detail, the alphabetic order of variables in table A2 is not completed, for instance Reynolds numbers and Prandl number. Therefore difficult to find.

———————————————

---

## Author Comment (AC2) · 23 Sep 2016

**First, we would like to thank Stefan Dekker for his kind and motivating words, as well as for his insightful questions and helpful suggestions. Below, we respond to the various comments one-by-one.**

i) Abstract: L9: you report two errors in the PM, without telling which ones (leaf exchanges of sensible heat and longwave radiation). Please add them, a reader would like to know what is missing.

ii) Also in the abstract I would like to see a statement that the error can be enormous.

For instance, in Fig 7,8 the authors report differences in modelled vs observed ET and H of up to 100 W/m2

**Good points, the sentence in the abstract will be modified to:**

**"Detailed analysis of the derivation by Monteith (1965) and later amendments revealed two errors, one in neglecting two-sided exchange of sensible heat by a planar leaf, and the other related to the representation of hypostomatous leaves, which are very common in temperate climates. Furthermore, we found that the neglect of feedbacks between leaf temperature and radiative energy exchange also contributes to bias in simulated latent heat flux by the PM equation, which was as high as 50% of the observed flux in some experiments."**

iii) Minor point: Figure 1, you write the Eq Rs=El+Hl+Rll, while in the text you write Rs=Rll+Hl+El

**Thanks, we will make this more consistent.**

iv) There are some inconsistencies in reporting units in the main text. I mean all units are correct but are not reported. I would be in favour that all variables if introduced for the first time should have a unit. For instance, no unit is given for Rs, while El (L71) does have a unit.

**Very good point, we will add units everywhere.**

v) L. 80-84, This part is introduced to fast. I do not understand this sentence. . . you introduce hc here (convective transport coefficient) What do you mean with leaf boundary conductance (the leaf boundary layer conductance to sensible heat is gsw (ant not hc)

**We apologise for the confusion caused by inconsistent terminology. We called $h_c$ the "convective heat transfer coefficient" above and here we called it the "leaf boundary layer conductance to sensible heat". We will make terminology more consistent and clarify that both $h_c$ and $g_{bw}$ relate to the same physical principles of diffusion and boundary layer dynamics.**

vi) I think it would be good to end chapter 2 with a new section (2.5) on the comparison of the different models (Rlin, MuC,PM, MU) and that you finally want to compare numerical solutions with PM and your new derived analytical solution (Rlin). This means that the part from the results L258-264, which are actually methods, should be incorporated in this new section (2.5). It will help the reader what he/she can expect.

**We will add a section 2.5, "Comparisons of numerical and analytical models with observations", leading into the description of the experimental setup. However, we will keep a condensed version of L258-264 at the beginning of the results section, as these sentences are meant to introduce the reader to the plots that follow.**

vii) Make new section 4.1: I think that the results should be divided in two parts. First 4.1 in which the data are compared to numerical solutions. In that sense I also would like to incorporate the data wind speed against H and ET measurements and numerical solutions. In this section it would be good to link back to Figure 1, were they show that H, ET are dependent on wind speed and Rs and Rll not.

**We will follow the above suggestion and in addition to Fig. 6, we will add experimental and numerical model results of the response of LE, H and Rnet to wind speed. We discuss this in more detail in the upcoming technical note, but the reviewer is right that this would be helpful in the present paper already.**

viii) Make new section 4.2: In section 4.2 you then can compare the different models. If possible I would like to see 4 graphs (35 pores/mm2 against wind speed, 7 pores/mm2 against wind speed, 35 pores/mm2 against VPD, 7 pores/mm2 against VPD). If they are not measured then only the 35 and 7 against VPD and 35 against wind speed. In this section I hope that the authors can tell a little bit more on the general behaviour of the models and why there is a clear order of over/under estimation depending on the models used. The authors could do that by also showing the resulted Rll ( (Eq2) versus Eq. (23) and as a result also the difference in Tl for the different models. Then I hope it becomes more clear why the PM, MUc,MU over and/or underestimate.

**We will follow the above suggestion and add a plot of 35 pores/mm2 against vapour pressure for comparison with 7 pores/mm2 against vapour pressure. Unfortunately, we have not conducted experiments with 7 pores/mm2 against wind speed in darkness, but we could add plots of 1.8 pores/mm2 or without a perforated foil, i.e. infinite conductance, if this is deemed necessary to increase confidence in the results. We tried to minimise the number of figures per amount of information, and with the added figure in Section 4.1. and one more in 4.2, we will be already at 10 figures in the main document. All additional plots will be included in the technical note. A plot of Eq. 2 against Eq. 23 for $R_{ll}$ was presented in Fig. A3. The effect of neglecting the $R_{ll}$-$T_l$ feedback is expressed in the difference between the Rlin and MUc results, while the PM result shows the result of added neglect of two-sided sensible heat flux and the even stronger under-estimation of LE by the MU equation results from further reducing latent heat flux in an attempt to account for one-sided transpiration on top of the erronously one-sided sensible heat flux. We will clarify this further in the revised manuscript.**

ix) Minor point: Results (L257). Here we only report two experiments under varying vapour pressure. This is true for Figure 6, but not for Figure 7 as here you also report

an effect of different wind speeds. Please add that.

**Thank you for pointing out this inconsistency. We will adapt the text to reflect the plots presented.**

x) Minor point: I am not in favour of saying that most of the results will be presented in a technical note

**Good point, we will reformulate to state:**

**"The ranges of stomatal geomtetries and deduced conductances for the two different leaves presented here are given in Table 1. A more detailed analysis of correspondence between experimental results obtained for a larger variety of artificial leaves and the numerical model will be presented in a technical note (Schymanski and Or, in prep.). Here, we only present selected experiments that highlight systematic differences between the various analytical solutions, the numerical model and observations."**

xi) L 288, again two errors without clearly telling which ones.

xii) L 292: The discussion directly continues in observations which are not shown, while I think that the authors could discuss the observations which are presented. Especially why some models over/under estimate the results is important to address and important for the readers of HESS

**Lines 288-297 will be modified to:**

**"In our mathematical analysis, we found two errors in the PM equation and in the "corrected" MU-formulation by Monteith and Unsworth (2013). Both formulations are based on evaporation from a soil surface, which exchanges sensible and radiative heat only on one side, whereas planar leaves have two sides**

exposed to the surrounding air. Failure to recognise this error led to a second error in the MU formulation, where an additional reduction to transpiration was introduced to represent leaves that exchange water vapour only on one side. For a leaf, the energy for transpiration in darkness is mainly supplied by sensible heat flux (on both sides), which increases with increasing wind speed. In contrast, the energy for evaporation from a soil surface in darkness is supplied by sensible heat on the evaporating surface only, (and by soil heat flux from below). The neglect of the additional uptake of sensible heat on the second side of the leaf in the PM and MU models led to significant under-estimation of the observed transpiration rates in our experiments. Note, however, that the bias is not constant and not always negative. As illustrated in Fig. 8, the negative bias decreases with increasing irradiance or air temperature, goes to 0 at a certain combination of temperature and irradiance and then becomes positive at higher values of irradiance and/or temperature. This is because under conditions when the leaf temperature is lower than ambient, sensible heat flux is a source of energy for transpiration, whereas under conditions when the leaf is warmer than the air, sensible heat flux competes with transpiration for energy. The omission of sensible heat exchange by the second leaf surface has therefore most drastic effects when leaf temperature most strongly deviates from air temperature. It may also be noteworthy in this context that the expression for aerodynamic resistance..."

Appendix: It is a real pleasure that, as far I can see, all equations and units are correct. One minor detail, the alphabetic order of variables in table A2 is not completed, for instance Reynolds numbers and Prandl number. Therefore difficult to find.

**Wow, thank you! This was due to different working names for the dimensionless numbers, which were used for sorting. This will be corrected. We will also make all the code and data available online, for easy verification and further use by the**

[Figure]

**reader. The preparation of the data and worksheets is in progress and can be accessed here: https://github.com/schymans/Schymanski_leaf-scale_2016**

---

## Referee Comment (RC2) · Anonymous Referee #2 · 20 Oct 2016

Comments on "Leaf scale experiments reveal important omission in the Penman-Monteith equation" by Schymanski and Or

My first feeling is that the title of the paper does not reflect its real content and that its content is not really appropriate for a hydrology journal such as HESS.

Despite an important theoretical development with 83 equations (27 in the main text + 56 in appendices), the title seems to induce that the "supposed" omission in the Penman-Monteith equation was revealed through experimental data. It is not true. The authors, in fact, derive an equation for the evaporation from a single leaf (Eq. 22) using a lot of mathematical details and then test the equation by means of an experimental setup. The theory precedes the experiment and justifies the experiment. Additionally,

the Penman-Monteith (PM) equation commonly refers to canopy evaporation and not to single leaf evaporation (e.g., ET0 in FAO Irrigation and Drainage Paper 56). The PM equation represents a particular form of the so-called combination equation, first derived by Penman (1948) for open water and then extended by Penman and Monteith to any evaporating surface (bare soil, leaf, canopy, etc ..). Speaking of PM equation at leaf scale can be somewhat misleading from my standpoint; it would be more appropriate to speak of combination equation.

Many aspects of the theoretical development, however, are not new and can be found in many textbooks or previous articles. The question of single leaf evaporation in relation with stomata distribution is an old issue. It has been addressed by many authors other than Monteith and Unsworth (for instance: Jarvis and McNaughton, 1986; Verhoef and Allen, 2000; Lhomme et al., 2012) and the question should be considered as closed from my standpoint. Assuming that the content of the paper is novel and relevant, HESS is certainly not the appropriate journal for such a topic. Plant, Cell and Environment or Journal of Experimental Botany should be more suitable. I should recognize, however, that the authors made a remarkable experiment in a wind tunnel with artificial leaves connected to a water supply, performing laser perforations and measuring all the components of the energy balance.

As far as I understand, the main point of the theory is the derivation of Eq. 22, which gives the evaporation from a single leaf (amphistomatous or hypostomatous) in the form of a combination equation (combining surface energy balance and convective transfers with the surrounding air). It is opposed to the so-called MU equation (Eq. 21), previously derived by Monteith and Unsworth in their reference book (Principles of Environmental Physics).The authors' equation (Eq. 22) appears to be correct, provided resistances ra and rs are defined as one-sided leaf resistances (this point, however, is not clear in the text: see P7 Line 156, where we could understand they are defined as two-sided). The authors claim that the MU equation, correct for amphistomatous leaves, is not correct for hypostomatous leaves because of a factor 2 missing in the

definition of the resistance ra in the nominator. I have checked Eq. 21 in the reference book of Monteith and Unsworth (P188 of the second edition 1990). Their demonstration is not perfectly clear because they do not give the complete combination equation for a single leaf; they only specify the change (their equation 11.30) in the denominator of the equation. One may suppose, nevertheless, that their equation is valid for amphistomatous leaves, but not for hypostomatous leaves. I must emphasize that by no means, the point mentioned above should be considered as an "important omission in the Penman-Monteith equation": first, because it has been correctly addressed in previous articles (those mentioned above), second and more importantly, because the authors do not assess the possible impact this "new" leaf formulation (and the small error supposedly encountered in the combination equation) can generate on the PM equation at canopy scale (the relevant scale for the hydrological community). It is the main problem of the paper.

I should add, as minor comment, that the beginning of the discussion section (leaf temperature and wind speed) is not clear and quite confusing insofar as it deals with "observations not presented here" (I quote).

References

Jarvis, A. J. and McNaughton, K. G.: Stomatal Control of Transpiration: Scaling Up from Leaf to Region, Advances in Ecological Research, 15, 1–49, 1986.

Verhoef, A. and Allen, S.J.: A SVAT scheme describing energy and CO2 fluxes for multi-component vegetation: calibration and test for a Sahelian savannah. Ecol. Model. 127, 245–267, 2000.

Lhomme, J.P., Montes, C., Jacob, F., Prévot, L.: Evaporation from heterogeneous and sparse canopies: on the formulations related to multi-source representations. Boundary-Layer Meteorology 144, 243-262, 2012.

---

## Author Comment (AC3) · 26 Oct 2016

Stanislaus J. Schymanski and Dani Or

stanislaus.schymanski@env.ethz.ch

**We thank the reviewer for clearly formulating his/her concerns about our manuscript and for giving us the opportunity to clarify a few issues that could be misleading. Below, we respond to the comments one-by-one.**

i) My first feeling is that the title of the paper does not reflect its real content and that its content is not really appropriate for a hydrology journal such as HESS.

**We are confident that our responses to the detailed comments below illustrate the adequacy of the title and the appropriateness of the manuscript for HESS.**

ii) Despite an important theoretical development with 83 equations (27 in the main text + 56 in appendices), the title seems to induce that the "supposed" omission in the Penman-Monteith equation was revealed through experimental data. It is not true. The authors, in fact, derive an equation for the evaporation from a single leaf (Eq. 22) using a lot of mathematical details and then test the equation by means of an experimental setup. The theory precedes the experiment and justifies the experiment.

**Although we do not see the relevance of the order of events (after all, this is not a mystery novel), we wonder about the basis for the reviewer's assertion that "it is not true". In fact, we confirm that the omission in the Penman-Monteith (PM) equation was revealed experimentally first. The experimental setup was devised to test the detailed leaf energy balance model we used in previous publications (Schymanski et al., 2013; Schymanski and Or, 2015, 2016), and the idea to compare results with the PM equation arose during the evaluation of the experimental data. Therefore, we believe that it is appropriate to state in the title that the experimental evidence revealed a problem, which was subsequently identified in the derivations.**

iii) Additionally, the Penman-Monteith (PM) equation commonly refers to canopy evaporation and not to single leaf evaporation (e.g., ET0 in FAO Irrigation and Drainage Paper 56). The PM equation represents a particular form of the so-called combination equation, first de- rived by Penman (1948) for open water and then extended by Penman and Monteith to any evaporating surface (bare soil, leaf, canopy, etc ..). Speaking of PM equation at leaf scale can be somewhat misleading from my standpoint; it would be more appropriate to speak of combination equation.

**The notion that the PM equation "refers to canopy evaporation and not to single leaf evaporation" is a common confusion in the scientific community. On Lines 22–28 in our manuscript, we suggest that the use of the PM equation at the**

**canopy scale may be the reason for employing various empirical corrections rather than testing the adequacy of the equation itself. In the revised manuscript, we will point out more clearly that Monteith (1965) referred to a single leaf when deriving the PM equation, as evident in the abstract of his paper and from Page 208 onwards. Use of the PM equation at canopy scale is commonly motivated in the context of a big leaf analogy, implying that the physics valid for a leaf are also valid for a canopy (e.g. Lhomme et al., 2012, cited by the referee below). Our point is that a physically-based equation should at least represent all relevant processes adequately at the scale of derivation before being upscaled. Therefore, the failure of the PM equation to reproduce fluxes at the leaf scale has to be considered potentially relevant for its performance at the canopy scale. Note that use of the term "combination equation", as proposed by the referee, would equally refer to the Penman and the Penman-Monteith equation, whereas we specifically focus on the PM equation, which was formulated for a leaf, as opposed to a wet soil surface, and has added consideration of stomatal resistance.**

iv) Many aspects of the theoretical development, however, are not new and can be found in many textbooks or previous articles. The question of single leaf evaporation in relation with stomata distribution is an old issue. It has been addressed by many authors other than Monteith and Unsworth (for instance: Jarvis and McNaughton, 1986; Verhoef and Allen, 2000; Lhomme et al., 2012) and the question should be considered as closed from my standpoint.

**In the present manuscript, we aimed to provide consistent and complete derivations, focusing on the key papers related to the original derivations of the PM equation and the interpretation considered as "textbook knowledge", i.e. the book by Monteith and Unsworth (2013). The referee is correct that Jarvis and McNaughton (1986) included a correction in the PM equation for amphistoma-**

tous leaves in the appendix, Eq. A9. However, they mistakenly explained the difference "as a result of our use of conductances defined on a single surface area basis", and did not alert the reader to the more fundamental issue we found, namely the missing half of sensible heat exchange. This is explained in our manuscript in Lines 170–174. We cannot see why the referee cites Verhoef and Allen (2000) and Lhomme et al. (2012) as examples for the treatment of single leaf evaporation. The former focuses solely on canopy evaporation and the latter just re-uses the formulation by Monteith and Unsworth (2013). To our knowledge, a general formulation as presented in our manuscript and its test for a single leaf has not been presented in the literature and hence we do not agree that the question should be considered as closed. On the contrary, our experimental evidence suggests that the question of the physical basis of the PM equation should be re-opened and re-evaluated not only at the leaf scale, but also at canopy scale.

v) Assuming that the content of the paper is novel and relevant, HESS is certainly not the appropriate journal for such a topic. Plant, Cell and Environment or Journal of Experimental Botany should be more suitable. I should recognize, however, that the authors made a remarkable experiment in a wind tunnel with artificial leaves connected to a water supply, performing laser perforations and measuring all the components of the energy balance.

This assessment by Referee 2 is in contrast to the assessment by Referee 1, and likely based on the misunderstanding that the PM equation was derived at canopy scale, while our correction is merely relevant to its application at the leaf scale. Closer study of the literature reveals that the PM equation is actually rarely used at the leaf scale, in favour of explicit solution of the leaf energy balance, e.g. Ball et al. (1988). In the revised manuscript, we will emphasise more clearly that the paper focuses on the physical fundamentals of a formula-

**tion central to hydrology, from the leaf to the canopy and continents, and that there is no additional physics involved in the scaling up of the PM equation from leaf to canopy other than ad-hoc upscaling from leaf to canopy resistance. Following up on the reviewer's statement at the beginning of the review, that the PM equation is mainly used at the canopy scale, we conclude that its physical basis is therefore of great relevance to Hydrology and Earth System Sciences.**

(vi) As far as I understand, the main point of the theory is the derivation of Eq. 22, which gives the evaporation from a single leaf (amphistomatous or hypostomatous) in the form of a combination equation (combining surface energy balance and convective transfers with the surrounding air). It is opposed to the so-called MU equation (Eq. 21), previously derived by Monteith and Unsworth in their reference book (Principles of Environmental Physics).The authors' equation (Eq. 22) appears to be correct, provided resistances ra and rs are defined as one-sided leaf resistances (this point, however, is not clear in the text: see P7 Line 156, where we could understand they are defined as two-sided).

**Actually, we consider Eqs. 25–27 as the main point of our theory, delivering general analytical solutions for latent heat flux, sensible heat flux and leaf temperature, while also considering the longwave radiative feedback. However, we acknowledge the referee's comment that a skimming reader might be confused by Lines 151–159, where we explore various alternative assumptions about the resistances to show that neither of them makes the PM equation physically consistent for a planar leaf. To clarify this, we will introduce this paragraph with the following sentence:**

**"To test whether Eq. 19 is physically consistent for a planar leaf, we will attempt to deduce it from our generally valid Eq. 10, using any suitable definitions for $c_E$, $c_H$, $r_a$ and $r_s$."**

vii) The authors claim that the MU equation, correct for amphistomatous leaves, is not correct for hypostomatous leaves because of a factor 2 missing in the definition of the resistance ra in the nominator. I have checked Eq. 21 in the reference book of Monteith and Unsworth (P188 of the second edition 1990). Their demonstra- tion is not perfectly clear because they do not give the complete combination equation for a single leaf; they only specify the change (their equation 11.30) in the denominator of the equation. One may suppose, nevertheless, that their equation is valid for amphistomatous leaves, but not for hypostomatous leaves

**We do not agree with the referee that the MU equation (Eq. 21 in our manuscript) is valid for amphistomatous leaves. In Line 171, we clarified that the MU equation is missing a factor of 2 in the denominator, representing two-sided exchange of sensible heat. This factor is independent of stomata being present on one or both sides of the leaf and hence makes the MU equation *invalid for any planar leaf* in a free air stream. As mentioned above and in our manuscript, some corrections can be found in the literature, but a systematic and explicit general derivation of the correct equations, as presented here, has been missing.**

viii) I must emphasize that by no means, the point mentioned above should be con- sidered as an "important omission in the Penman-Monteith equation": first, because it has been correctly addressed in previous articles (those mentioned above), second and more importantly, because the authors do not assess the possible impact this "new" leaf formulation (and the small error supposedly encountered in the combination equation) can generate on the PM equation at canopy scale (the relevant scale for the hydrological community). It is the main problem of the paper.

**Here, the referee seems to dispute that the omission we identified persists in the literature and additionally questions its relevance for canopy-scale process**

**representation. We hope that we have established clearly that the PM and MU equations indeed omit an energy balance term (that cannot be recovered by redefining resistances) and that this omission has not been rectified in the literature, as claimed by the referee. Given that the PM equation is commonly scaled up to the canopy by considering the canopy as a "big leaf", the effects of the omission seen at the leaf scale are likely similarly important in canopy-scale models. We have not yet extended the analysis to canopy scale response and thus cannot comment on the propagation of the omission to larger scales and how much of the effect of the "small error" may be eliminated or obscured by canopy-scale parameterisation. However, we are confident that the reviewer is not suggesting to overlook simple errors in the basic representation, in the hope these will not affect larger scales? Moreover, we have shown that the sensitivity of ET to future temperature changes is likely represented incorrectly by the PM equation, as its derivative remains anchored in the erroneous leaf level representation.**

ix) I should add, as minor comment, that the beginning of the discussion section (leaf temperature and wind speed) is not clear and quite confusing insofar as it deals with "observations not presented here" (I quote).

**Thank you for pointing this out. The discussion of the leaf temperature feedback on Lines 291–297 actually detracts from our main point and will be replaced by the following:**

**"This results in a strong under-estimation of latent heat flux by the PM equation in our experiments, where sensible heat flux is the main source of energy for evaporation (in the absence of shortwave radiation). "**

**References**

Ball, M., Cowan, I., and Farquhar, G.: Maintenance of Leaf Temperature and the Optimisation of Carbon Gain in Relation to Water Loss in a Tropical Mangrove Forest, Functional Plant Biol., 15, 263–276, http://www.publish.csiro.au/paper/PP9880263, 1988.

Jarvis, A. J. and McNaughton, K. G.: Stomatal Control of Transpiration: Scaling Up from Leaf to Region, Advances in Ecological Research, 15, 1–49, 1986.

Lhomme, J. P., Montes, C., Jacob, F., and Prévot, L.: Evaporation from Heterogeneous and Sparse Canopies: On the Formulations Related to Multi-Source Representations, Boundary-Layer Meteorology, 144, 243–262, doi:10.1007/s10546-012-9713-x, http://link.springer.com/article/10.1007/s10546-012-9713-x, 2012.

Monteith, J. L.: Evaporation and environment, Symposia of the Society for Experimental Biology, 19, 205–234, http://www.ncbi.nlm.nih.gov/pubmed/5321565, 1965.

Monteith, J. L. and Unsworth, M. H.: Principles of environmental physics: plants, animals, and the atmosphere, Elsevier/Academic Press, Amsterdam ; Boston, 4th edn., 2013.

Schymanski, S. J. and Or, D.: Wind effects on leaf transpiration challenge the concept of "potential evaporation", Proceedings of the International Association of Hydrological Sciences, 371, 99–107, doi:10.5194/piahs-371-99-2015, http://www.proc-iahs.net/371/99/2015/, 2015.

Schymanski, S. J. and Or, D.: Wind increases leaf water use efficiency, Plant, Cell & Environment, 39, 1448–1459, doi:10.1111/pce.12700, http://doi.wiley.com/10.1111/pce.12700, 2016.

Schymanski, S. J., Or, D., and Zwieniecki, M.: Stomatal Control and Leaf Thermal and Hydraulic Capacitances under Rapid Environmental Fluctuations, PLoS ONE, 8, e54 231, doi:10.1371/journal.pone.0054231, http://dx.doi.org/10.1371/journal.pone.0054231, 2013.

Verhoef, A. and Allen, S. J.: A SVAT scheme describing energy and CO2 fluxes for multi-component vegetation: calibration and test for a Sahelian savannah, Ecological Modelling, 127, 245–267, doi:10.1016/S0304-3800(99)00213-6, http://www.sciencedirect.com/science/article/pii/S0304380099002136, 2000.

---

## Author Response (AR1)

**Description of changes in revised manuscript**

Stan Schymanski and Dani Or

December 4, 2016

Dear editor, dear reviewers,

We would like to thank you again for all the insightful comments and for giving us the opportunity to improve our manuscript and submit a revised version. We implemented all the changes we promised in our original responses to the reviewers and added a few additional improvements, as described below. In the below summary of changes, we refer to the attached version of the manuscript with highlighted changes (note the different line numbers compared to the revised manuscript itself). We also wish to point out that the accompanying technical note, containing the experimental details and additional results, has been submitted to HESS as manuscript hess-2016-643.

We hope that the revised manuscript satisfies the high quality standards of HESS and look forward to your response.

**1 Changes to document structure**

Following the suggestions by Stefan Dekker, we slightly re-structured the manuscript in the following way.

- In the Methods, we added a Section 2.5, "Comparisons of numerical and analytical models with observations"

- The results section was subdivided into two sub-sections, one focusing on the correspondence between experimental results and the numerical model, and the other on the performance of the different analytical solutions.

**2 Added content**

Apart from adding Section 2.5 and 3.1, following the suggestions by Stefan Dekker, we expanded Figs. 6 and 7, providing additional examples as well as metrics of energy balance closure and the role of longwave emission in the experimental data. We also added units for each variable in the text as requested, and also reviewed our table of symbols (Table A1) to be more consistent and complete. In response to both reviewers, we added additional text

to clarify the importance of the findings and their relevance for the hydrological community (abstract, L9–13, introduction, L31–36, and in the discussion, L356–366, L379–385). We also updated the online code and data, enabling readers to reproduce our figures and re-use the code for additional analysis: `https://github.com/schymans/Schymanski_leaf-scale_2016.git`.

**3 Other improvements**

As suggested by both reviewers, we have removed bits of text that had little relevance for the present paper and sharpened the message in various places. We have also made additional measurements of the perforations and updated Table 1 to reflect the new and more systematically analysed values. The analysis itself is described in detail in the recently submitted companion paper, hess-2016-643.

[revised manuscript text omitted]